# Discovery of immunotherapy targets for pediatric solid and brain tumors by exon-level expression

Timothy I. Shaw [1,2,11], Jessica Wagner [3,11], Liqing Tian [1,3,11], Elizabeth Wickman [3,4], Suresh Poudel[5], Jian Wang[1], Robin Paul [1], Selene C. Koo [6], Meifen Lu[6], Heather Sheppard[6], Yiping Fan[7], Francis H. O'Neill [8], Ching C. Lau[8,9,10], Xin Zhou [1], Jinghui Zhang [1,12] ✉ & Stephen Gottschalk [3,12] ✉

Immunotherapy with chimeric antigen receptor T cells for pediatric solid and brain tumors is constrained by available targetable antigens. Cancer-specific exons present a promising reservoir of targets; however, these have not been explored and validated systematically in a pan-cancer fashion. To identify cancer specific exon targets, here we analyze 1532 RNA-seq datasets from 16 types of pediatric solid and brain tumors for comparison with normal tissues using a newly developed workflow. We find 2933 exons in 157 genes encoding proteins of the surfaceome or matrisome with high cancer specificity either at the gene ($n = 148$) or the alternatively spliced isoform ($n = 9$) level. Expression of selected alternatively spliced targets, including the EDB domain of fibronectin 1, and gene targets, such as COL11A1, are validated in pediatric patient derived xenograft tumors. We generate T cells expressing chimeric antigen receptors specific for the EDB domain or COL11A1 and demonstrate that these have antitumor activity. The full target list, explorable via an interactive web portal (https://cseminer.stjude.org/), provides a rich resource for developing immunotherapy of pediatric solid and brain tumors using gene or AS targets with high expression specificity in cancer.

Immunotherapy with T cells expressing chimeric antigen receptors (CARs) holds the promise to improve outcomes for pediatric solid tumors, including brain tumors[1]. However, in contrast to CAR T-cell therapy for CD19-positive hematological malignancies, the antitumor activity of CAR T cells for solid and brain tumors has been limited[2].

Lack of efficacy is most likely multifactorial, including limited T-cell fitness, inefficient homing to tumor sites, the hostile tumor microenvironment, and a limited array of targetable antigens[2,3].

The majority of approaches for identifying new surfaceome targets for pediatric solid tumors have largely relied on differential gene

[1]Department of Computational Biology, St. Jude Children's Research Hospital, Memphis, TN 38105, USA. [2]Department of Biostatistics and Bioinformatics, H. Lee Moffitt Cancer Center and Research Institute, Tampa, FL 33612, USA. [3]Department of Bone Marrow Transplantation and Cellular Therapy, St. Jude Children's Research Hospital, Memphis, TN 38105, USA. [4]Graduate School of Biomedical Sciences, St. Jude Children's Research Hospital, Memphis, TN 38105, USA. [5]Center for Proteomics and Metabolomics, St. Jude Children's Research Hospital, Memphis, TN 38105, USA. [6]Department of Pathology, St. Jude Children's Research Hospital, Memphis, TN 38105, USA. [7]Center for Applied Bioinformatics, St. Jude Children's Research Hospital, Memphis, TN 38105, USA. [8]The Jackson Laboratory for Genomic Medicine, Farmington, CT 06032, USA. [9]Connecticut Children's Medical Center, Hartford, CT 06106, USA. [10]University of Connecticut School of Medicine, Farmington, CT 06032, USA. [11]These authors contributed equally: Timothy I. Shaw, Jessica Wagner, Liqing Tian. [12]These authors jointly supervised this work: Jinghui Zhang, Stephen Gottschalk. ✉e-mail: jinghui.zhang@stjude.org; stephen.gottschalk@stjude.org

expression analysis[4], often in a specific cancer type[5–7]. These approaches may lead to missed opportunities for finding pan-cancer targets that are effective across multiple types, some of which may arise from alternative splicing. We are interested in identifying cancer-specific exons (CSEs) herein referring to those with high expression in tumor but restricted and limited expression in normal tissues; some of which encode tumor-associated antigens which can serve as CAR targets[8].

To discover CSE targets for immunotherapy for pediatric solid and brain tumors, we analyze 1532 RNA-seq samples from the St. Jude/Washington University Pediatric Cancer Genome Project (PCGP), the National Cancer Institute (NCI) Therapeutically Applicable Research to Generate Effective Treatment (TARGET), and St. Jude Children's Research Hospital (St. Jude) Cloud[9] real-time clinical genomics data (ClinGen). Using these data sets, we identify a total of 2933 cancer-specific exons in 157 genes encoding surfaceome or matrisome, including 9 alternatively spliced (AS) isoform targets. We validate several targets in cell lines, patient-derived xenograft models (PDXs) and primary tumors, and demonstrate that CAR T cells specific for two identified targets have antitumor activity against pediatric sarcoma.

## Results

### Discovery of CSE in pediatric solid and brain tumors
To identify CSEs (Fig. 1A), we analyzed the cancer-specific transcription profiles of RNA-seq data from 840 solid and 692 brain tumor samples (Fig. 1B). Major types of solid tumors included (i) adrenocortical carcinoma (ACC, $n = 22$), (ii) desmoplastic round cell tumor (DSRCT, $n = 9$), (iii) Ewing sarcoma (EWS, $n = 20$), (iv) melanoma (MEL, $n = 31$), (v) neuroblastoma (NBL, $n = 219$), (vi) osteosarcoma (OS, $n = 136$), (vii) retinoblastoma (RB, $n = 23$), (viii) rhabdomyosarcoma (RMS, $n = 86$), (ix) Wilms tumor (WT, $n = 158$), (x) other solid tumors (other ST, $n = 136$). Major types of brain tumor include (i) choroid plexus carcinoma (CPC, $n = 21$), (ii) ependymoma (EPN, $n = 139$), (iii) high grade glioma (HGG, $n = 155$), (iv) low grade glioma (LGG, $n = 140$), (v) medulloblastoma (MB, $n = 126$), and (vi) other brain tumors (other BT, $n = 111$). For normal tissue comparison, we analyzed 7460 RNA-seq samples across 30 normal tissues from the Genotype-Tissue Expression (GTEx) database (Supplementary Fig. 1).

CSEs were identified by an analytical pipeline involving the following five main steps (Fig. 1C): (1) map RNA-seq data of 1532 tumor samples and 7460 normal tissue samples, (2) select cancer-specific exons based on enriched expression in tumors, (3) retain exons that are present in proteins present on the cell surface (surfaceome) or extracellular matrix (ECM; matrisome)[10–12], (4) curate targets based on expression specificity in cancer, and (5) classify CSEs according to aberrant gene-level transcription or AS isoforms in tumors.

Our CSE pipeline resulted in the identification of 67,472 exons in 2273 genes, which were enriched in tumors compared to normal tissues. Of these, 3964 exons in 249 genes belonged to the surfaceome or matrisome. We further classified these into Tier 1 and Tier 2 targets (Fig. 2; Supplementary Figs. 2–4; Supplementary Data 1) with Tier 1 targets having minimal expression in matching normal tissue types and vital organs such as brain, liver, and lung. Gene-level Tier 1 targets were additionally required to have low expression in normal bone marrow samples using a logistic regression model that we previously developed[13]. To ensure Tier 1 targets having high and low protein abundance in tumor and normal tissues, respectively, we further analyzed PDX and GTEx proteomics data resulting in 37 Tier 1 and 120 Tier 2 targets.

We identified 16 AS in 9 genes (7 genes with 1 AS, 1 gene with 2 AS, and 1 gene with 7 AS) (Supplementary Data 1). For the two genes in which we identified >1 AS, we selected the most differential AS for the final CSE list, which included 9 AS (5 [Tier 1]; 4 [Tier 2]) and 148 gene-level targets (Fig. 1C). Forty-two CSEs were present in the matrisome and surfaceome (AS: 1 [Tier 1], 2 [Tier 2]; gene-level: 13 [Tier 1], 26

[Tier 2]), 68 only in the surfaceome (AS: 2 [Tier 2]; gene-level: 12 [Tier 1], 54 [Tier 2]), and 47 only in the matrisome (AS: 4 [Tier 1]; gene-level: 7 [Tier 1], 36 [Tier 2]). Protein expression of Tier 1 and 2 targets was confirmed using published proteomic data sets[14,15]. To assess whether expression of the 9 AS targets was associated with variants at the splice sites, we analyzed 504 samples with matched tumor WGS data in our pediatric cancer cohort and did not find any significant association in two genes (FN1, VCAN) that harbored such variants (Supplementary Data 2). No such variants were found in the other seven genes. This indicates that the AS targets were caused by transcription deregulation instead of genomic variants in pediatric cancer samples that we analyzed.

### Landscape of CSE immunotherapeutic targets in pediatric cancers
Tier 1 and 2 targets identified by our analysis encoded proteins with diverse biological functions such as cell adhesion, collagen, ECM, receptor, and signaling factors (Fig. 2; Supplementary Figs. 2–4). Interestingly, 56.7% (89 out of 157) of the targets were associated with the matrisome, indicating that ECM proteins provide a rich source of tumor-specific antigens. Of the 9 AS targets, three (FN1, TNC, and COL6A3) showed high expression in OS, and all three were confirmed by full-length transcriptome sequencing of 3 OS patient samples (Supplementary Fig. 5).

To our knowledge, amongst the 157 targets, 11 (CD83, CD276 (B7-H3), FAP, FN1, GPC2, GPC3, IL1RAP, KDR, KIT, MET, PROM1 (CD133)) have been explored as CAR targets in preclinical studies[5,8,16–26], while the remaining 146 (93%) are novel. Of the known targets, 5 (FAP, B7-H3, GPC3, KDR, CD133) have been or are actively being explored in clinical studies. Six known oncofetal proteins were also on the target list, which includes AS isoforms of FN1 and TNC as well as gene-level targets TGFB2, WNT5A, GPC3 and IGF2. Given their limited expression in normal tissue beyond the fetal development stage, these oncofetal proteins should be considered high priority targets. Similarly, testis-restricted targets such as SPA17, TEX14, LAMA1, SMOC1, TNFAIP6, GPC2, and COL20A1 (Fig. 2; Supplementary Fig. 3) may also be leveraged for their cancer-specificity. Indeed, GPC2-CAR T cells have already been developed for NBL[5,27].

Approximately 30% (49 of 157) of the targets are highly expressed in both solid and brain tumors at high prevalence (≥25%) in at least one tumor type (Supplementary Figs. 2, 4), highlighting the potential for developing pan-cancer targets. These include 6 of the 9 AS targets (FN1, TNC, NRCAM, PICALM, FYN, VCAN). Specifically, FN1 encodes fibronectin which is involved in cell adhesion and migration processes including embryogenesis, wound healing, blood coagulation, host defense, and metastasis[28]. The identified FN1 AS target encodes the alternatively spliced extra domain B of FN1 (EDB)[29] which is highly expressed in all solid and brain tumor types except RB, with the highest prevalence in OS, EWS, RMS, WT, MEL, HGG and EPN (Supplementary Fig. 2). TNC encodes an extracellular matrix protein that plays a role during normal development, including neural migration, as well as tumorigenesis[30,31]. The TNC AS target encodes the alternatively spliced C domain of TNC[32] with high expression detected at high prevalence in HGG, EPN, OS and MEL (Supplementary Fig. 2). VCAN is a member of the aggrecan/versican proteoglycan family and is involved in cell adhesion, proliferation, proliferation, migration and angiogenesis. Mutations in VCAN can cause Wagner syndrome type 1[33]. The VCAN AS target encodes VCAN isoform 1 (VCAN1), which has the highest expression levels in LGG, HGG, and DSRCT (Supplementary Fig. 2). Pan-cancer gene-level targets include procollagen 11A1 (COL11A1), which encodes one of the two alpha chains of type XI collagen and is expressed at high prevalence in OS and CPC (Supplementary Fig. 2). Mutations in COL11A1 are associated with type II Stickler syndrome and with Marshall syndrome[34]. GPC3 which is highly expressed in RMS, WT, and

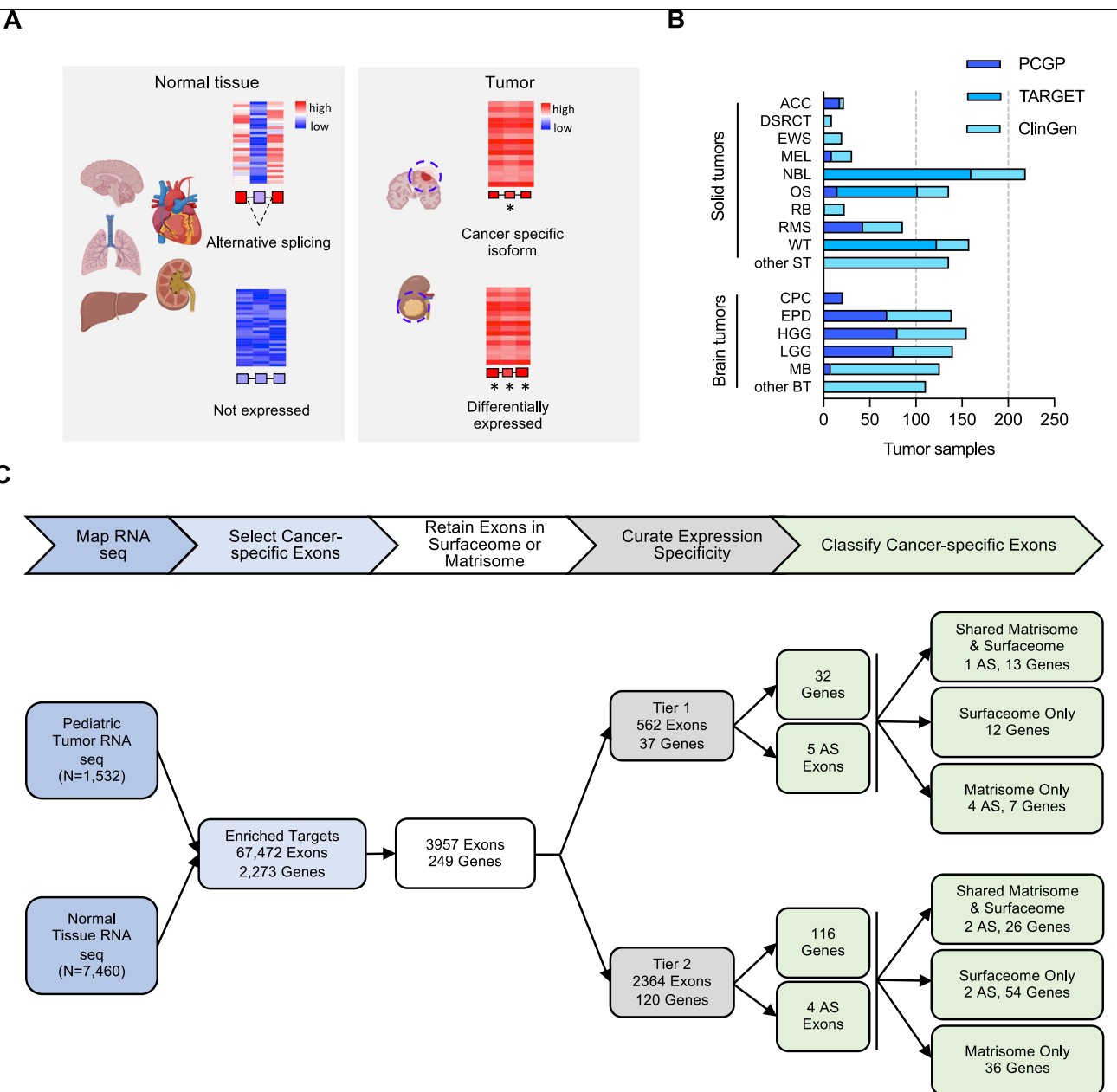

**Fig. 1 | Discovery of CSEs as immunotherapy targets in pediatric solid and brain tumors. A** Schematic illustration of the concept of exploiting CSEs, which include both gene-level and alternatively spliced exons as targets for immunotherapy (in part created with BioRender software). **B** Pediatric solid (*n* = 840) and brain (*n* = 692) tumor RNA-seq data sets used for discovery. The sample count in each tumor type is colored by the data source (i.e., PCGP, TARGET, and St. Jude's ClinGen). Nine major types of solid tumors are marked by their abbreviations as follows: adrenocortical carcinoma (ACC, *n* = 22), desmoplastic round cell tumor (DSRCT, *n* = 9), Ewing sarcoma (EWS, *n* = 20), melanoma (MEL, *n* = 31), neuroblastoma (NBL, *n* = 219), osteosarcoma (OS, *n* = 136), retinoblastoma (RB, *n* = 23), rhabdomyosarcoma (RMS, *n* = 86), and Wilms tumor (WT, *n* = 158). Rare solid tumors are binned into the category of other solid tumors (other ST, *n* = 136). Five

brain tumor types are shown by their abbreviation as follows: choroid plexus carcinoma (CPC, *n* = 21), ependymoma (EPN, *n* = 139), high-grade glioma (HGG, n = 155), low-grade glioma (LGG, *n* = 140), medulloblastoma (MB, *n* = 126). Rare brain tumors are binned into other brain tumors (other BT, *n* = 111). **C** Analysis workflow (top) and resulting data (bottom) involving the following steps: (1) Quantify exon-level expression by RNA-seq mapping; (2) Select exons highly expressed in tumor but not normal tissues; (3) Retain exons from surfaceome/matrisome; (4) Perform curation expression specificity to remove artifacts and to categorize Tier 1 and Tier 2 candidates representing those without and with expression in adjacent/critical tissues (e.g., brain, liver, bone marrow); Tier 1 targets also require to have low proteomics expression in GTEx. (5) Classify targets into AS exons versus gene-level.

CPC (Supplementary Fig. 4A, B), is a member of glypican family and regulates the signaling of WNTs, Hedgehogs, fibroblast growth factors, and bone morphogenetic proteins. Loss of function mutations in GPC3 can cause Simpson-Golabi-Behmel syndrome[35]. CD276 which is highly expressed in ACC, OS, WT, and HGG (Supplementary Fig. 4A, B) belongs to the immunoglobulin superfamily and regulates T-cell-mediated immune responses[36,37].

## Validation of cell surface expression of selected CSEs in patient-derived xenograft (PDX) models

Validation was carried out on three Tier 1 targets, FN1, VCAN1, COL11A1, which are expressed in a broad spectrum of pediatric brain and solid tumors based on our analysis. For VCAN1 and EDB, we took advantage of mAbs that recognize the part of the molecule that is encoded by the differentially expressed exon and performed flow

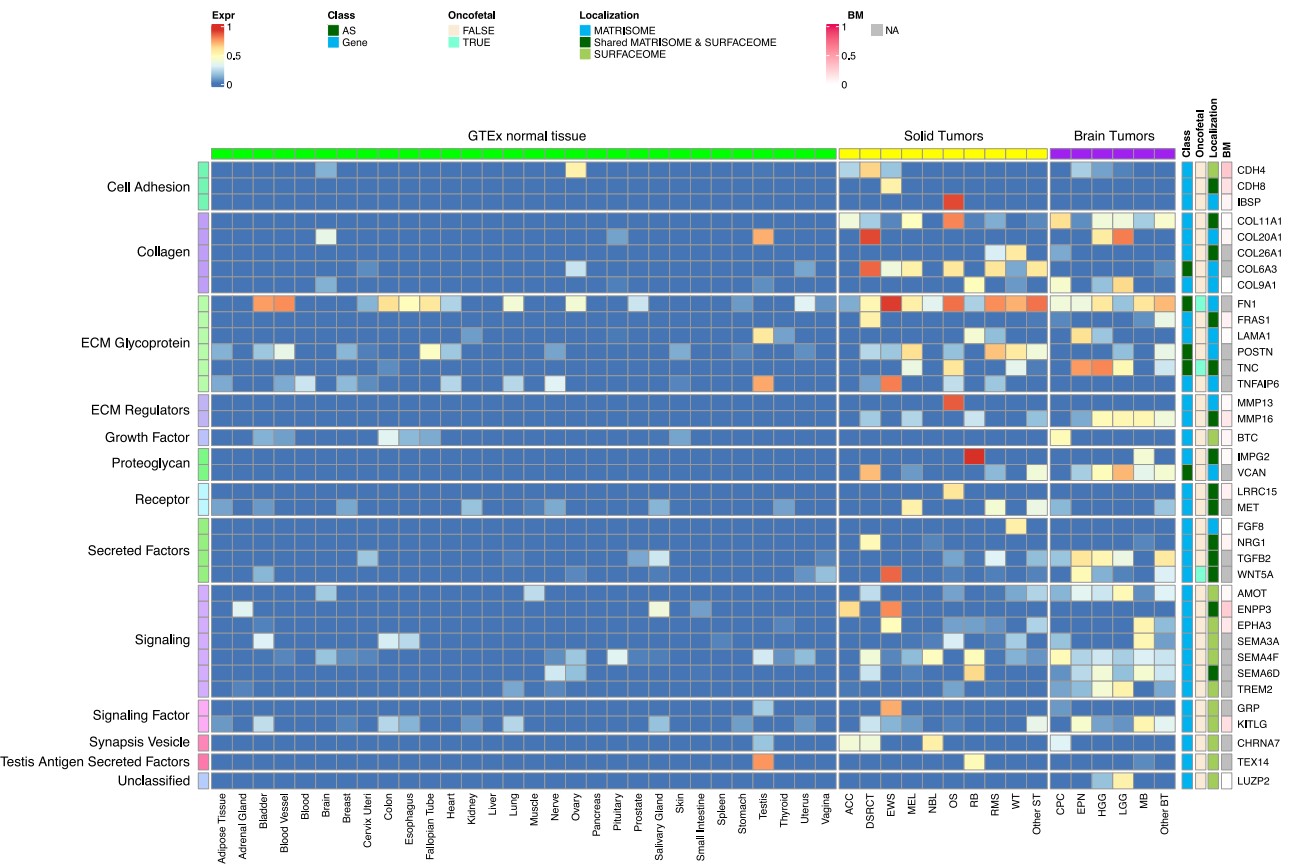

**Fig. 2 | Normalized expression of Tier 1 targets across pediatric cancer types and normal tissues.** The heatmap uses a blue-red color scale to display the mean expression rank (range 0–1) of exon FPKM value of RNA-seq samples profiled for a normal tissue (*n* = 7460) or a specific tumor type (*n* = 1532). 37 Tier 1 targets are grouped by their biological functions on the left, while the gene names, status on AS exon, oncofetal protein, cellular localization, and expression prevalence in normal bone marrow are shown on the right. One representative exon is shown for a gene-level target. Expr expression, BM bone marrow, NA not available.

cytometric analysis of 15 pediatric PDX samples (5 OS, 5 EWS, 5 RMS; Supplementary Data 3). VCAN1, EDB, and COL11A1 were expressed in >50% of tumor cells in 8 or 9/15 PDX samples (Fig. 3A–D). In addition, for COL11A1, we performed immunohistochemistry (IHC) on the PDX samples as well as primary tumor samples. For the PDX sample, there was concordance between flow cytometric and IHC analysis with only one flow + /IHC- tumors (Supplementary Fig. 6). Of the primary tumors, 12/18 OS, 7/11 EWS, and 14/37 RMS samples highly expressed COL11A1 (Supplementary Fig. 7); 5/18 OS, 3/11 EWS, and 10/37 RMS tumor samples showed low expression, respectively. All stained normal tissues remained negative (Supplementary Fig. 8). We confirmed EDB and COL11A1 expression in all tumors (100%) by RT-qPCR in 12/12 PDX samples analyzed (4/4 OS, 4/4 EWS, 4/4 RMS) (Fig. 3E). Finally, we evaluated COL11A1 expression using publicly available single-cell RNA-seq data generated from 11 tumor samples[38] and confirmed its presence in 10 out of 11 tumor samples (Supplementary Fig. 9).

### COL11A1-CAR and EDB-CAR T cells have antitumor activity against multiple types of pediatric sarcoma

We focused on developing a CAR T-cell therapy approach for COL11A1. In addition, we extended our previous study, in which we had demonstrated that T cells expressing a functional CAR with an EDB-specific single chain variable fragment (scFv) antigen binding domain[39] (EDB-CAR T cells), recognize and kill one OS (LM7) and one EWS (A673) cell line[19]. We generated a COL11A1.CD28.z-CAR (COL11A1-CAR) with a COL11A1-specific scFv derived from the 1E8.33 mAb, which was raised against a peptide sequence that is unique for COL11A1[40]

(Supplementary Fig. 10A, B). COL11A1- and EDB-CAR T cells were generated by retroviral transduction and expression were confirmed by flow cytometry (Fig. 3F; Supplementary Fig. 10C, D). We performed 48-h co-culture assays with COL11A1-positive pediatric tumor cell lines (OS: LM7, 143B; RMS: CCL-136, CRL-2061; EWS: A673) and COL11A1-negative primary fibroblasts (Fig. 3G).

COL11A1- and EDB-CAR T cells produced significant amounts of IFNγ in comparison to NT T cells only in the presence of antigen-positive tumor cells (Fig. 3H). Likewise, both CAR T-cell populations had significant cytolytic activity against antigen-positive tumor cells in comparison to NT T cells in a standard cytotoxicity assay, confirming specificity (Fig. 3I). To confirm that the newly generated COL11A1-CAR is antigen-specific, we performed additional orthogonal assays. COL11A1-CAR T cells did not recognize 143B cells in which COL11A1 was knocked out (KO) by CRISPR/Cas9 gene editing, and T cells expressing a non-functional COL11A1-CAR with mutated immunoreceptor tyrosine-based activation motifs (ITAMs) did not kill wildtype 143B cells (Supplementary Fig. 11).

In the final set of experiments, we evaluated the antitumor activity of COL11A1-CAR T cells in vivo. We first utilized our established osteosarcoma model where LM7.GFP.ffLuc cells were injected intraperitoneally (i.p.) into NSG mice followed by one single i.v. dose of 3×10⁶ COL11A1-CAR or NT T cells on day 7 (Fig. 4A)[41]. COL11A1-CAR T cells had significant anti-tumor activity as judged by bioluminescence imaging in 10/10 mice in comparison to NT T cells, which had no antitumor activity (Fig. 4B, C). This resulted in significant median survival advantage of >100 days post COL11A1-CAR T-cell infusion (Fig. 4D), and surviving mice had no clinical evidence

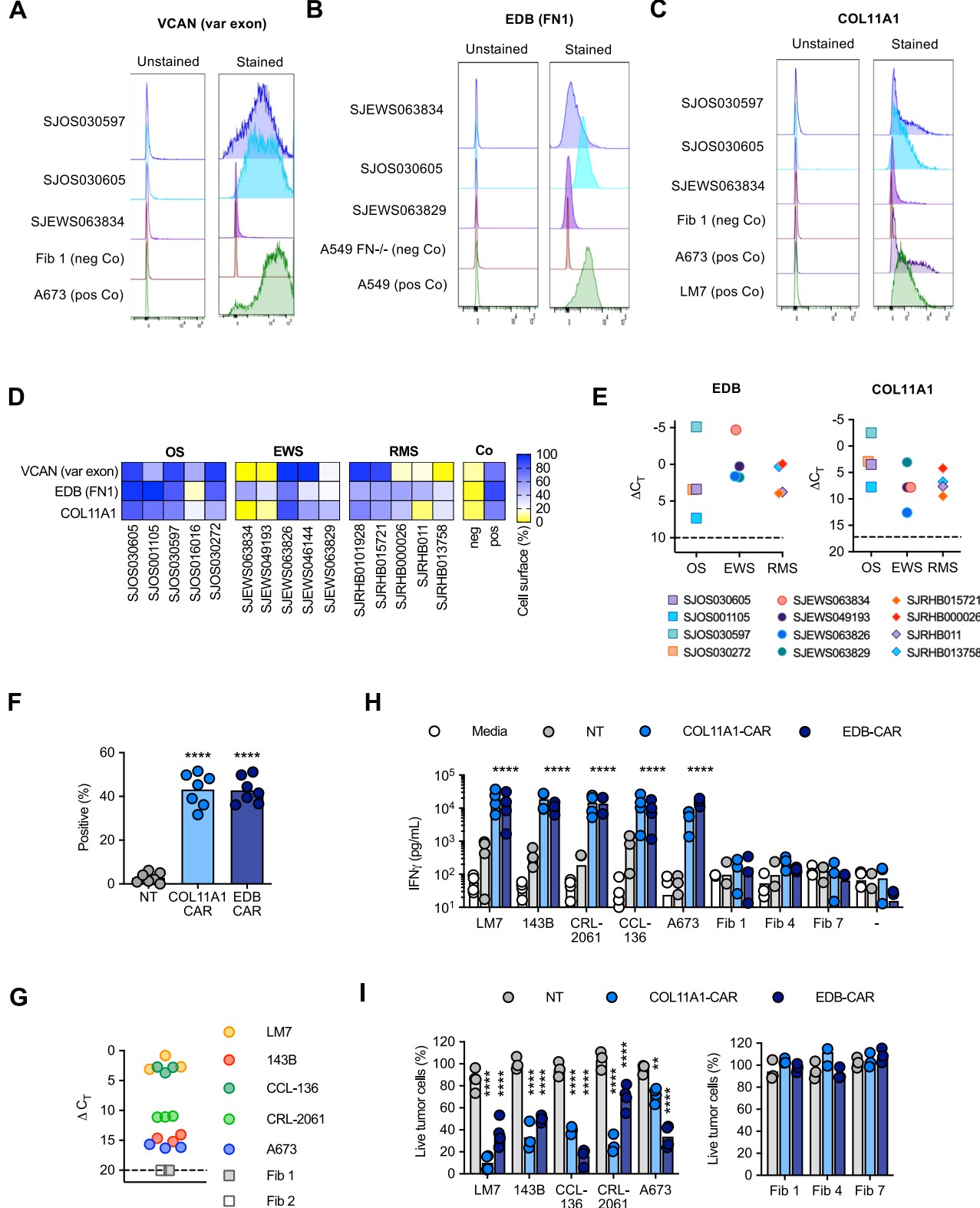

of xenogeneic graft versus host disease as judged by inspection of their fur coat and absence of weight loss (Fig. 4E). Since tumors eventually recurred, we explored mechanisms of tumor recurrence using the same model with LM7 cells and GFP.ffLuc-expressing CAR or NT T cells (Supplementary Fig. 12A). We observed CAR T cell expansion but limited persistence, and decreased expression of COL11A1 on day 65 post tumor cell injection (Supplementary

Fig. 12B–E), indicating that tumor recurrence is most likely due to both mechanisms.

We confirmed the antitumor activity of COL11A1-CAR T cells using our subcutaneous EWS (A673) model[41] in which tumor-bearing mice received one single i.v. dose of $1 \times 10^6$ COL11A1-CAR or NT T cells on day 7 (Fig. 4F). COL11A1-CAR T cells had robust antitumor activity, resulting in a significant survival advantage in comparison to NT T cells (Fig. 4G-I).

**Fig. 3 | COL11A1-CAR T cells have potent antitumor activity in vitro.**
**A−C** Representative histogram of cell surface expression of 3 out of 5 PDX samples for VCAN (variable (var) exon), FN1 (EDB), and COL11A1. Tumor cell lines (A673, A549, LM7) served as positive controls (pos Co) and normal fibroblasts (Fib 1) or A549 FN-/- as negative controls (neg Co). **D** Heat map displaying cell surface expression in PDX samples as determined by flow cytometry ($n = 5$ per tumor type). OS osteosarcoma, EWS Ewing's sarcoma, RMS rhabdomyosarcoma, Co control, Fib normal fibroblast ($n = 1$ per target, average of 3 technical replicates, negative and positive controls same as panel 3**A**−**C**). **E** RTqPCR for EDB or COL11A1 gene expression performed on PDX samples. Delta CT calculated relative to GAPDH. Dashed line: threshold of positivity based on qPCR results of antigen negative cells. **F** CAR expression determined by flow cytometry using an anti-mouse IgG F(ab')2 ($n = 7$ biologically independent samples, two-way ANOVA, ****$p < 0.0001$). **G** COL11A1 expression of LM7, 143B, CCL-136, CRL-2061, and A673 tumor cells, and

two primary fibroblast cell lines (Fib 1, Fib 2) determined by RT-qPCR. Triplicates for each cell line are shown; COL11A1 expression in fibroblast was undetectable and their ΔCT value was set as 20. **H** NT or COL11A1-CAR T-cells were incubated at a 2:1 E:T ratio for 48 h with tumor cells or primary fibroblasts. Media only samples served as controls. IFNγ in culture media was determined by ELISA ($n = 3$) for Fib 1, 4, and 7, $n = 4$ for all other conditions, biologically independent donors, two-way ANOVA of log-transformed data comparing against the NT of each tumor type to COL11A1 of each tumor type; ****$p < 0.0001$. All fibroblast experiments were non-significant. **I** Cytolytic activity of NT or COL11A1-CAR T-cells at an E:T ratio of 4:1 for 72 h against GFP.ffluc-expressing tumor cells or primary fibroblasts ($n = 3$ for 143B, CCL-136, CRL-2061 (COL11A1-CAR and NT), and Fib 1, 4, 7, $n = 4$ for all other conditions, biologically independent donors, two-way ANOVA, comparing against the NT of each tumor type to COL11A1 of each tumor type **$p = 0.0058$, ****$p < 0.0001$). All fibroblast experiments were non-significant.

## Exploring CSE targets on the CSE-miner data portal

We developed a web-based data portal, CSE-miner (https://cseminer. stjude.org/) to enable biomedical researchers to explore all targets identified in this study. The data portal includes rich visualization features to allow evaluation of omics data used for CSE identification, along with ancillary information useful for designing future experiments. To illustrate the functionality of the data portal, we used the EDB exon of FN1 as an example. Each target can be explored using four different views as follows: (1) A pan-target scatter plot for prioritizing targets based on the relative expression of tumor samples and normal tissues (Fig. 5A); (2) a table view for selecting a CSE of interest to examine its expression pattern across all tumor types and normal tissues (Fig. 5B); (3) a heatmap view showing the relative expression in tumor and normal samples across all exons within the gene, highlighting identified CSE targets (Fig. 5C); and (4) a gene view which can toggle between a genome view highlighting the specific exons, and a protein view highlighting the domains encoded by the identified targets, along with examples of associated antibody binding regions and proteome expression using mass spectrometry data from CPTAC pediatric brain tumors[14] and St. Jude's RMS xenograft tumors[15] (Fig. 5D).

The visualization features implemented in CSEminer were designed to support target prioritization, which requires verifying high-level expression in tumor types and limited expression in normal tissues. This is facilitated by a box plot of normalized expression values and a bar graph of quartile distribution across tumor types and normal tissue types implemented in the table view. Additionally, the gene view enables distinguishing a gene-level target from an AS-exon target, while additional information (e.g., mAb availability) helps with planning future experiments. We illustrated these visualization features for three additional examples that were evaluated for selection of high priority targets: VCAN, COL11A1, and TNC (Supplementary Figs. 13–15).

## Discussion

In this study, we describe a pan-cancer analysis for discovery of CSEs as potential targets for immunotherapy for pediatric solid and brain tumors. Using the large RNA-seq datasets generated by multiple genomic initiatives, we identified 157 gene-level or alternatively spliced exon targets encoding members of surfaceome or matrisome. These targets were further categorized into Tier 1 ($n = 37$) or Tier 2 ($n = 120$), requiring that Tier 1 candidates show minimal expression in matching normal tissue types and vital organs. To our knowledge, the majority (93%) of identified targets were novel. Previously identified targets included CD276 and GPC3, and CAR T cells targeting these antigens have been evaluated in early phase clinical studies with an encouraging safety profile[42-44]. However, we classified these targets as Tier 2 targets based on expression in vital organs. This highlights that gene expression not necessarily correlates with protein expression[45]. Likewise, antigen density is critical for efficient target cell recognition by CAR T cells[46]. Thus, Tier 2 targets should not be a priori excluded, but

require additional studies to further assess the risk of on-target/off-cancer toxicity. The employed algorithm to identify targets might have detected membrane associated proteins that are not expressed on the cell surface, and additional validation studies have to be conducted for individual targets. Of note, we believe that these proteins should not be excluded a priori, since mislocalization of proteins have been described in cancer[47].

In the present study, we used the normal tissue expression from GTEx as a control for identifying CSE targets in pediatric cancer. A potential caveat of this approach is that GTEx samples were from adult tissues, which may not completely match the normal expression in children. An ongoing public initiative, the developmental GTEx project aimed at stablishing a molecular and data analysis resource for gene expression in multiple relatively healthy reference neonatal, pediatric, and adolescent tissues, may ultimately provide a more accurate normal control for the childhood cancer cohort (https://www.genome. gov/Funded-Programs-Projects/Developmental-Genotype-Tissue-Expression). Currently, finding an appropriate match to the normal developmental stage of a pediatric cancer type remains extremely challenging as reactivation of fetal oncoprotein and immature developmental processes have thus far revealed critical therapeutic vulnerabilities for developing immunotherapy or small molecule-based interference for childhood cancer[48]. For example, antibodies against the fetal antigen GD2, which is expressed by neuroblastoma, are now routinely used in the treatment of high-risk neuroblastoma, and GD2-CAR T cells have also shown promising results in early phase clinical studies[49,50]. The vast majority of the CSE targets we identified were due to aberrant expression at the gene level as only 9 targets (COL6A3, FN1, POSTN, TNC, VCAN, NRCAM, FYN, PICALM and CLSTN1) were due to alternative splicing. This may be related to the use of exons defined by Gencode v31 gene models, which limits our ability to find AS targets in novel isoforms. Future studies that incorporate novel isoform discovery with CSE analysis or other newly published methods such as Isoform peptides from RNA splicing for Immunotherapy target Screening (IRIS)[51] followed by validation using proteomics databases may further expand the repertoire of AS targets. Our studies demonstrated that FN1 and COL11A1, targets that are associated with the matrisome, are expressed by cancer cells. For adult cancers, these are also expressed by stromal and/or endothelial cells of the tumor microenvironment (TME)[52,53], and additional studies are needed to investigate this for pediatric cancers.

Our analysis focuses on identifying CSEs as candidate immunotherapeutic targets themselves rather than on peptides derived from these exons that are presented by MHC molecules[54]. We decided on this approach so that the candidate targets can be broadly recognized by CAR T cells. In contrast, HLA-restricted peptides can, in general, only be targeted with MHC-restricted, αβ T-cell receptor (TCR) T cells[55], although antibody based approaches are also being developed[56]. The CSE targets we identified are in genes with diverse biological functions. In some cases, gene-level overexpression in

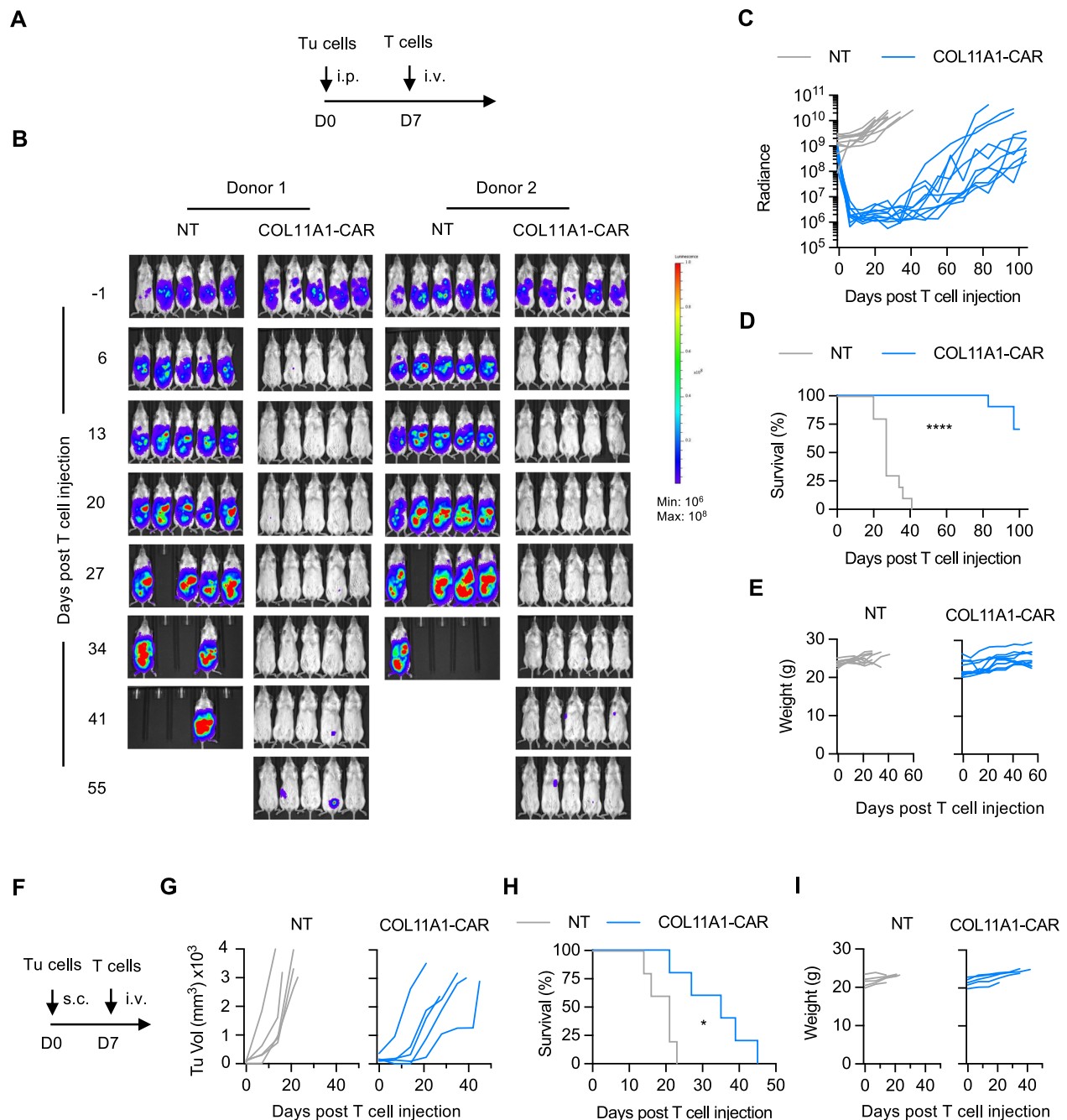

**Fig. 4 | COL11A1-CAR T cells have potent antitumor activity in vivo. A** Schematic of LM7 OS animal experiment. Day 0: i.p. injection of $1 \times 10^6$ LM7.GFP.ffluc cells; Day 7: i.v. injection of $3 \times 10^6$ NT or CAR T-cells ($n = 10$) mice per group; 2 different biologically independent T-cell donors (5 mice per donor); Tu: tumor cells; intraperitoneal (i.p.), subcutaneous (s.c.), intravenous (i.v.) injection. **B** Representative bioluminescence images until day 55 post tumor cell injection (all life mice are shown for each group ($n = 10$ mice per group)). **C** Quantitative bioluminescence data (Radiance: photons/sec/cm²/sr) ($n = 10$ mice per group). **D** Kaplan-Meier survival curve, log-rank (Mantel-Cox) test; ****$p < 0.0001$ ($n = 10$ mice per group). **E** Weight (g) of mice ($n = 10$ mice per group). (**F**) Schematic of A673 EWS animal experiment. Day 0: s.c. injection of $1 \times 10^6$ A673 cells; Day 7: i.v. injection of $3 \times 10^6$ NT or CAR T-cells ($n = 5$ mice per group). (**G**) Tumors measured weekly by caliper measurements ($n = 5$ mice per group). (**H**) Kaplan-Meier survival curve, log-rank (Mantel-Cox) test, *$p = 0.0112$ ($n = 5$ mice per group). **I** Weight (g) of mice.

pediatric cancer is known without the knowledge of the expressed isoform. For example, TNC, a glycoprotein, is known to be highly expressed in pediatric EPN and HGGs[57,58]. Our exon-based analysis also identified several splice variants (e.g., C domain of TNC, EDB, COL6A3) known to be enriched in adult cancers[29,32,40]. This has broad therapeutic implications for pediatric cancers, since exon-targeted immunotherapies or imaging approaches that are currently being developed for adult cancer could be readily applied to pediatric cancer. Additional CSEs have been reported for FN1 and TNC[28,30]. While we

excluded the extradomain A of FN1 as an CSE due to expression in several normal tissues (Supplementary Fig. 16A), we identified additional CSEs for TNC, which have been reported in adult cancer[30], including the CSE that encode the D domain of TNC (Supplementary Fig. 16B).

We took advantage of PDX models of common pediatric solid tumors (OS, EWS, RMS) to quantify the expression of VCAN1, COL11A1, and EDB. Using orthogonal assays such as flow cytometry, IHC, and RT-qPCR, we were able to consistently detect the expression of these

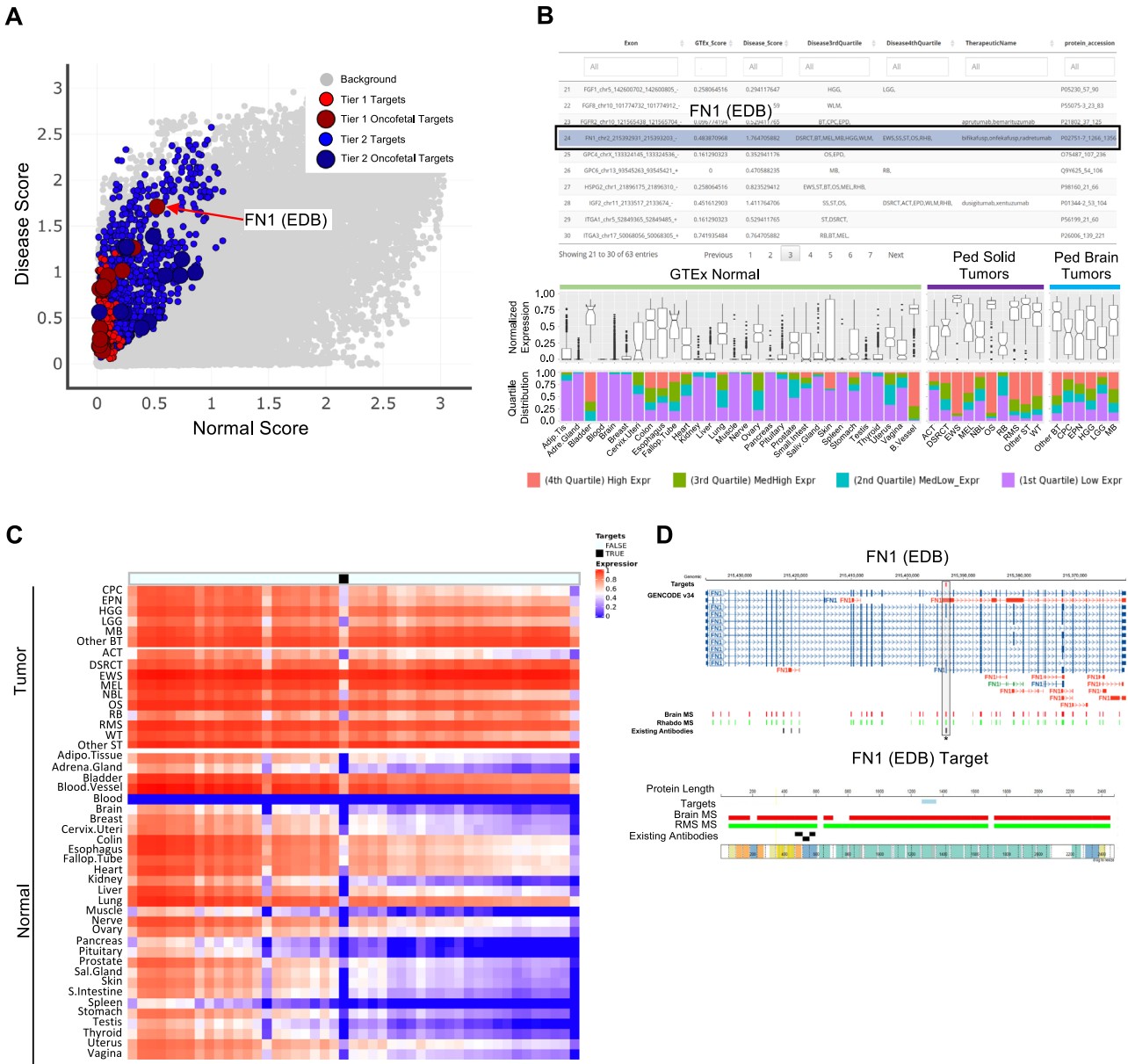

**Fig. 5 | Navigation and visualization with the CSE Miner web portal.** An alternatively spliced exon in FN1 encoding the FNB isoform is highlighted to illustrate the visualization features. On the portal (https://cseminer.stjude.org/), users can explore Tier 1 and Tier 2 candidates through the (**A**) pan-target view, (**B**) table view, (**C**) heatmap view, and (**D**) gene view. **A** All candidates can be visualized through a two-dimensional scatter plot showing the normalized mean exon expression (details in Methods) across pediatric tumor samples (*x*-axis) and GTEx normal samples (*y*-axis). **B** On the table view, the distribution of exon expression in normal tissues (*n* = 7460) and tumor types can be examined in percentile (middle panel) and quartiles (bottom panel). In the notched box plot, the lower, middle and upper hinges of the box plots correspond to the 25th, 50th and 75th percentiles, respectively. The notch displays the 95% confidence interval of the median. The upper whisker of the box plot extends from the upper hinge to the largest value no further than 1.5 × IQR from the upper hinge. IQR, interquartile range or distance between 25th and 75th percentiles. The lower whisker extends from the lower hinge to the smallest value at most 1.5 × IQR from the lower hinge. Data beyond the end of the whiskers are outlier points. **C** Heatmap showing exon expression across normal tissues (*n* = 7460) and tumor (*n* = 1532) types within a candidate gene. A user can select a transcript of interest (the transcript ENST00000432072.6 was selected for FN1) from the list shown in the left panel. Heatmap represents a mean normalized rank expression for each tissue type (right panel). **D** A gene view which can toggle between a genomic view highlighting the exons or a protein view displaying the relevant protein domains.

splice variants and gene expression, highlighting the robustness of our analytical approach. Although gene expression does not always correlate with protein expression, we found overall good correlation between our conducted assays.

We and other investigators had recently shown in preclinical models that EDB-CAR T-cells have potent antitumor activity, targeting not only tumor cells but also endothelial cells of the tumor vasculature[19,20]. These studies suggested that ECM proteins like FN1 that adhere to the cell surface can serve as CAR targets. To explore if this also applies to other ECM proteins, we generated CAR T cells specific for COL11A1, which was expressed at high levels in OS and CPC. COL11A1-CAR T cells recognized and killed COL11A1-positive tumor cells in vitro and had potent antitumor activity in vivo in two pediatric sarcoma xenograft models. While tumors eventually recurred, treated mice had a survival advantage. The survival advantage was particularly striking in our LM7 OS model, which expressed COL11A1 at high levels. We observed limited CAR T cell persistence and decreased expression of COL11A1 in recurring tumors, and future studies are required to gain

additional insight into the mechanism of recurrence and explore the therapeutic benefit of COL11A1-CAR T cells that are further genetically modified to enhance their effector function. Our finding that COL11A1 can serve as a CAR target should have broad implications since COL11A1 is also expressed in adult cancers with poor prognosis, such as pancreatic adenocarcinoma[59], and has been proposed as a novel biomarker[52].

In conclusion, by performing a comprehensive data mining using the rich RNA-seq data sets, we have demonstrated that the surfaceome/matrisome of pediatric solid and brain tumors contains cancer-specific exons that can serve as candidates for cancer immunotherapy. We identified and validated candidate targets with orthogonal assays and demonstrated that CAR T cells constructed from these targets have potent antitumor activity. Validation of consistent expression of target genes and to exclude epitope masking due to the tertiary structure of the protein in individual tumor cells is critical, which may involve performing IHC of primary patient samples and evaluating gene expression at single cell level by re-analyzing appropriate scRNA-seq data sets as demonstrated in our validation of COL11A1. While we focused here on CAR T cells, the identified antigen could serve as targets of mAbs, immunocytokines or antibody drug conjugates. The full data set, explorable online (https://cseminer.stjude.org/), provides a comprehensive roadmap for developing future immunotherapies in childhood cancer.

## Methods

### Ethical regulations
Blood from healthy donors was collected under an Institutional Review Board approved protocol at St. Jude Children's Research Hospital, after written informed consent was obtained in accordance with the Declaration of Helsinki. All animal experiments were conducted under a protocol approved by St. Jude Children's Research Hospital Institutional Animal Care and Use Committee.

### RNAseq data source
Solid and brain pediatric tumor RNA-seq data were downloaded from the St. Jude Cloud[9] (https://platform.stjude.cloud/data/cohorts/pediatric-cancer) for St. Jude/Washington University Pediatric Cancer Genome Project (PCGP) and St. Jude's Clinical Genomics (ClinGen) program. NCI TARGET data were downloaded from dbGaP under accession phs000218. RNA-seq data from the normal tissues were generated by the Genotype-Tissue Expression (GTEx) consortium[60] and downloaded from the GTEx portal (https://gtexportal.org release v7). All RNA-seq sample accession numbers are provided as Supplementary Data 4.

### RNAseq mapping and exon quantification
RNA-seq reads were mapped using the STAR 2.7.1a program in two-pass mode[61] to the human hg38 genome build using Gencode v31 primary assembly gene annotation gene models. Annotation of the exons status was based on APPRIS[62]. We used htseq[63] to quantify the exon level expression and converts gene transfer format (GTF) to exon-specific GTF. Specifically, we ran htseq-count using the parameters below to ensure reads with multiple mapping were incorporated when measuring expression of exons that have high-fidelity paralogous duplications (see Supplementary Fig. 17 for an example):

$$htseq-count-fbam-rpos-a0-sno-munion \\ -texon--nonunique\ all \quad (1)$$

If a read spans a splice junction, it would be counted for both exons, which could potentially lead to overestimation of expression level of a short exon. Read counts were further normalized to FPKM (fragments per kilobase of transcript per million mapped reads) and to mitigate the potential bias on short exons, we used read length

(instead of exon length) for normalizing exons that are shorter than the read length. The source code and documentation for each analysis can be found in GitHub (https://github.com/shawlab-moffitt/CSEminer-manuscript/tree/main/1_rnaseq_mapping_exonquant).

### Selection of cancer-specific exons (CSE) by performing tumor-vs-normal differential expression analysis
Differential expression was performed based on Wilcoxon rank-sum test. Let X1,...,Xn be the exon expression of tumor tissues, and Y1,...,Yn be the exon expression of normal tissue.

$$U = \sum_{i=1}^{n} \sum_{j=1}^{m} S(X_i, Y_j) \quad (2)$$

With

$$S(X,Y) = \begin{cases} 1, & if\ Y < X \\ \frac{1}{2}, & if\ Y = X \\ 0, & if\ Y > X \end{cases} \quad (3)$$

We then estimated the Z-score by a normal approximation of the U-statistics. Let n1 be the length of the number of samples from a cancer type, and let n2 be the number of samples from a normal tissue. ($m_U$ and $\sigma_U$) are the mean and standard deviation of U.

$$m_U = \frac{n1 n2}{2} \quad (4)$$

$$\sigma_U = \sqrt{\frac{n1 n2 (n1 + n2 + 1)}{12}} \quad (5)$$

$$z = \frac{U - m_U}{\sigma_U} \quad (6)$$

To provide a meta-comparison of consistently differentially expressed exons, we applied Stouffer's meta-analysis to combine k pairs of disease to normal comparison.

$$Composite\ Zscore \sim \frac{\sum_{i=1}^{k} w_i Z_i}{\sqrt{\sum_i^k w^2}} \quad (7)$$

with the weight defined as the median percentile rank differential between tumor and normal tissue.

$$w = median(Percentile\ Rank(X)) - median(Percentile\ Rank(Y)) \quad (8)$$

Solid tumors and brain tumors were analyzed separately. A candidate CSE exon is required to have a composite Z-score > 1 and above-median expression in at least one tumor type. To ensure low expression in normal tissues, candidates are also required to have ≤ 5 normal tissues expressed above the median level. We did not perform a gender analysis since there is no evidence that the underlying biology of childhood cancers is different in males and females.

### Protein annotation for CSE targets
We retained targets encoding surfaceome or matrisome based on the following data sets: The Cell Surface Protein Atlas[64], the MGI GO annotation[65], the human protein atlas[66], MatrixDB[67], and the compartment database[68]. We started by using Ensembl for the reference gene annotation which includes 59,088 genes and 226,950 transcripts. Genes that are tumor suppressors or known to be DNA binding, such as transcription factors and chromatin regulators, were filtered out. This resulted in 67,472 exons from 2273 genes encoding extracellular or

surfaceome proteins. The transmembrane information was predicted based on TMHMM server 2.0[10]. Intersecting this reference surfaceome or matrisome gene set with CSE candidates resulted in 249 genes encoding 3957 CSEs. Oncofetal annotation was derived from text mining from GeneCard followed by manual curation. Genes associated with tumor suppressors, transcription factor, epigenetic factors, kinases, cell differentiation factors, cytokine growth factors, and gene with the homeodomain were downloaded from MsigDB[69] (Supplementary Data 1). The status of 82 tumor suppressors in pediatric cancer were verified using mutation data on PeCan portal (https://pecan.stjude.org) which were curated from > 5000 pediatric cancer patients.

### Curation of expression specificity and splicing pattern of CSE targets

CSEs were characterized as either gene-level or AS exon targets based on the following criteria: (a) transcripts with < 40% CSE coverage were subjected to further examination as candidates; and (b) AS targets were required to match an alternatively spliced transcript in the reference database.

Candidate CSE targets for a cancer type profiled by multiple data resources (e.g. OS was profiled by ClinGen, PCGP and TARGET) required cross-validation of their expression in individual data resource to minimize the impact of coverage bias caused by different RNA-seq protocol. Additionally, verification of target expression in a proteomics database was required. In this study we used proteomics data generated from PDX models of pediatric solid tumors[15] and brain tumors[14] for this purpose. Additionally, any candidate targets identified in brain tumor which also exhibit high expression in normal brain (medium expression above $3^{rd}$ quartile) or a significant bias for exon position in GTEx data set (P-value < 0.01 for Pearson correlation between exon expression and exon number) are removed. The exon position bias check removes false positives caused by the 3′ bias in mRNA-seq protocol used by GTEx

Those targets that passed the QC check described above were further classified as Tier 1 or Tier 2 based on their expression status in normal tissues. Specifically, a Tier 1 candidate is expected to pass the following check: (1) absence of high expression in normal tissues paired to the tumor as follows: ACT/Adrenal, WLM/Kidney, RHB/Muscle, RB/Nerve, Mel/Skin; (2) low expression level in normal bone marrow samples for gene-level targets; and (3) low protein expression in normal tissues based on the GTEx proteomics data. Those that failed in any of these checks were classified as Tier 2. Details of analysis on normal bone marrow samples and GTEx Proteomics data are described below. Evaluation of expression level in normal bone marrow samples is needed because these normal samples were not profiled by GTEx. Apng the method described in reference 13[13] we determined the expression level of the target genes using data from reference 67[70]. As microarray data were generated for gene-level expression, we were not able to determine the expression status in bone marrow for AS targets.

To ensure that low protein expression of Tier 1 targets in normal tissues, we analyzed the GTEx proteomics data from http://gbsc-share.stanford.edu/GTEx_raw_files. We first normalized the peptide spectral matches (nPSM) to the exon length of the peptide-protein-sequence coded within each exon and identified a bimodal distribution of the nPSMs and used the optim function to identify a cutoff point separating the two modes (Supplementary Fig. 18). For each protein, we calculated an average nPSM based on the exon information and categorized candidates with high normal GTEx pro abundance which are subsequently downgraded to Tier 2.

### Tumor versus normal score for pan-cancer scatter plot

We generated an expression score for each exon to enable visual inspection of expression level in tumor versus normal for all candidate targets on the pan-target scatter plot. First, we calculated a binned score based on quartiles of exons that are above 1 FPKM. Exons below 1

FPKM were set to 0. We then calculated the mean of binned score for each tumor type and normal tissue type. Finally, we used the average of binned score across all tumor types and all normal tissue types to set the tumor score and normal score, respectively.

$$\text{Binned Score}(Xi) = \begin{cases} 0, \text{if quartile}(\text{median}(Xi)) \sim \text{"1st quartile"} \\ 1, \text{if quartile}(\text{median}(Xi)) \sim \text{"2nd quartile"} \\ 2, \text{if quartile}(\text{median}(Xi)) \sim \text{"3rd quartile"} \\ 3, \text{if quartile}(\text{median}(Xi)) \sim \text{"4th quartile"} \end{cases} \quad (9)$$

$$\text{Average Binned Score}(X) = \frac{\sum_{i=1}^{n} \text{Binned Score}(Xi)}{n} \quad (10)$$

n represents the number of tumor types or normal tissue types

### Validation of AS targets using full-length transcriptome sequencing data of OS patient samples

We generated libraries and performed Iso-Seq sequencing for 3 OS patient samples on a PacBio RSII instrument. The raw data files were processed according to the PacBio Isoseq3 pipeline which utilizes a number of command line tools provided in PacBio SMRT Tools v10.2 (https://www.pacb.com/support/software-downloads/). The pipeline generates non-redundant full-length (FL) transcripts in the following steps for each tumor sample: (i) compute consensus sequences and read quality, (ii) remove primers and adapters, (iii) remove polyA tail and artificial concatemers, (iii) de novo isoform-level clustering, (iv) minimap2 aligns FL transcripts to human reference (GENCODEv40), (v) transcripts were collapsed based on genomic mapping abundance was estimated and GTF annotation file generated, (vi) sqanti3 performed transcript classification and generated a reference corrected transcriptome fasta file. The GTF file was searched for transcripts matching the gene target region coordinates.

### Splice variant analysis

To determine whether the splice variants affect the expression of alternatively spliced exons identified in the 9 genes, we obtained genomic variants for 504 tumor samples which have their WGS data available on St Jude Cloud Genomic Platform (https://platform.stjude.cloud/). To find splice variants, we queried the tumor variant files which contain both somatic and germline variants for those located within 10 bp of splice acceptor and donor sites of the 9 AS exons in COL6A3 (chr2:237378636-237379235), FN1 (chr2:215392931-215393203), POSTN (chr13:37574572-37574652), TNC (chr9:115048260-115048532), VCAN(chr5:83519349-83522309), NRCAM (chr7:108191254-108191283), FYN (chr6:111699515-111699670), PICALM (chr11:85990250-85990378), and CLSTN1 (chr1:9756481-9756510). No variants were found for COL6A3, POSTN, TNC, NRCAM, FYN, PICALM, and CLSTN1. For the remaining two genes (FN1 and VCAN), no association between variants and expression level was detected based on 1-sided t-test, not surprising given the very low variant prevalence (1 out of 300 for FN1 and 21 out of 225 for VCAN) in tumors with median and high-level expression.

### Proteomics data analysis

To examine the protein-coding potential of candidate exons, we leveraged existing mass spectrometry profiling data sets generated from cancer cells relevant to our study. These included the deep mass spectrometry profiling of RMS[15], brain tumors[14], and patient derived xenograft (PDX) models were downloaded from the St. Jude proteomics facility and Clinical Proteomic Tumor Analysis Consortium (CPTAC). MS raw data were processed using the COMET software (http://comet-ms.sourceforge.net/)[71], an open-source fast MS/MS sequence database search tool using a fast cross-correlation

algorithm[72]. Briefly, raw MS files were searched against the human database downloaded from UniProt (52,490 entries) with Met oxidation as a dynamic modification. Search parameters were precursor and product ion mass tolerance (6 ppm and 10 ppm, respectively), fully tryptic restriction, two maximal missed cleavages, static TMT modification (+229.162932 Da on N-termini and Lys residues), dynamic Met oxidation (+15.99492 Da), three maximal dynamic modification sites, and the consideration of a, b, and y ions. Peptide-spectrum matches (PSMs) were filtered by seven amino acids minimal peptide length, mass accuracy (-3 ppm), and matching scores cutoff of < 2 xcorr and Δxcorr > 0.1. The visualization of the mass-spectrometry peaks was performed on the Msviewer[73].

## Tumor cell lines

143B (OS, CRL-8303), CRL-2061 and CCL-136 (RMS), and A673 (EWS, CRL-1598) cell lines were purchased from the American Type Tissue Collection (ATCC). The lung metastatic osteosarcoma cell line LM7 was kindly provided by Dr. Eugenie Kleinerman (MD Anderson Cancer Center, Houston, TX) in 2011. Primary fibroblast (Fib) cell lines from healthy donors were previously established[74]. The generation of all tumor cell lines expressing an enhanced green fluorescence protein firefly luciferase fusion gene (GFP.ffluc) was previously described[18]. The COL11A1 KO 143B cell line was generated by St. Jude's Center for Advanced Genome Engineering (CAGE) using CRISPR/Cas9 gene-editing technology. All cell lines were grown in DMEM or RPMI (Fisher Sci SH30022.01; Genclone 25-506 N) supplemented with 10% fetal bovine serum (FBS; GE Healthcare Life Sciences HyClone, SH3008803) and 2 mM Glutamax (Invitrogen, 35050061). Cell lines were authenticated using ATCC's human STR profiling cell authentication service every 6 months during the study. Cell lines were free of mycoplasma contamination, and routinely checked for Mycoplasma by the MycoAlert Mycoplasma Detection Kit monthly (Lonza, LT07-118).

## Patient-derived xenograft samples

Orthotopic patient-derived xenograft samples, collected under the Molecular Analysis of Solid Tumors (MAST) protocol, were provided by the Childhood Solid Tumor Network (CSTN) collection at St. Jude (https://cstn.stjude.cloud/search/)[75]. The gene expression from the primary patient tumors and PDX models are highly correlated ($R > 0.8$) except for patient samples with low tumor purity (purity <20%) due to the high-level admixture of gene expression in stromal cells (see Extended Data Fig. 3 of the CSTN manuscript)[75]. All samples were handled in accordance with CSTN policy including DNA profiling for short tandem repeat validation to confirm orthotopic (O)-PDX models between passages. Samples were hand homogenized in PBS (Lonza, 17-512 F) with 1% FBS (HyClone, SH3008803) and filtered twice through polystyrene test tubes with cell strainer caps (Falcon, 352235). Single cell suspension was used for both flow cytometry and real-time PCR.

## Primary sarcoma tissue sections

After St. Jude Institutional Review Board approval, deidentified archival formalin-fixed paraffin-embedded tissue blocks from clinical patient tumor samples were cut and H&E-stained sections were reviewed for correct diagnosis and tumor content by a pediatric pathologist (SCK). Matched unstained tumor sections were then stained. Samples were delineated into 3 categories based on expression levels: high, low, and negative based on normal tissue controls.

## Immunohistochemistry

To detect COL11A1 expression, IHC was performed on Ventana Discovery Ultra autostainer (Roche, Indianapolis, IN) with the following protocol and reagents. Vial of mAb anticol11a1 (Oncomatryx, High Concentration 2.3 mg/mL Rabbit monoclonal (Clone 1e8.33)). All reagents were provided by Roche, Indianapolis IN: Samples underwent heat-induced epitope retrieval, (Cell Conditioning Solution ULTRA CC1

(950-224, Roche)) for 32 min; the primary antibody was incubated for 30 min per manufacturers instruction; followed by DISCOVERY OmniMap anti-Rt HRP (760-4457; Roche), DISCOVERY ChromoMap DAB kit (760-159; Roche), Hematoxylin II (790-2208; Roche), and Bluing reagent (790-2037; Roche) were used for visualization. All samples (PDX, xenograft, primary, normal) were stained in along with positive control (LM7 xenograft) and negative control xenograft (143B COL11A1 KO xenograft), grown subcutaneously in NSG female mice, and tumors were harvested when they reached a size of 1000 mm³. Isotype controls were used as well.

## Reverse transcription quantitative PCR

mRNA extraction from single cell suspensions of cultured cell lines ($<1 \times 10^7$ cells) and PDX samples was performed using the Maxwell RSC simplyRNA Blood kit (Promega AS1380) on a Maxwell RSC machine. RT-qPCR was performed according to the manufacturer's instructions with 10 ng of RNA and 200 nM of primers using the Power SYBR Green RNA-to-$C_T$ 1-Step Kit (Thermo Fisher Scientific, 4389986) on an Applied Bioscience QuantStudio 6 Flex machine, and analyzed using QuantStudio software (Thermo Fisher Scientific). GAPDH primers were purchased from IDT (PrimeTime qPCR Primers, human GAPDH, Hs.PT.39a.22214836). Primers (IDT) were designed to detect EDB and COL11A1 using the NCBI Primer-BLAST tool.

EDB domain of FN1 Forward: 5′-CCC CAA CTC ACT GAC CTA AGC-3′

EDB domain of FN1 Reverse: 5′-CTG CCG CAA CTA CTG TGA TG-3′

COL11A1 Forward: 5′- CAG ACG GAG GCA AAC ATC GT-3′

COL11A1 Reverse: 5′-TCA TTT GTC CCA GAA ACA TGC C-3′

## Generation of retroviral vectors

In-fusion cloning (Takara Bio, 638947) was used to generate the COL11A1-CAR with a CD28 costimulatory domain and IgG1 short hinge using our retroviral vector as a template, which encodes a EphA2-CAR.CD28ζ expression cassette, a 2 A sequence, and truncated CD19[76]. The COL11A1-specific scFv was derived from mAb 1e8.33[40] and synthesized by GeneArt (Thermo Fisher Scientific). The non-functional COL11A1-CAR with mutated (mut) ITAMs was generated by using our retroviral vector encoding a CD28z.mut.CAR as a template[19]. The sequences of the final constructs were verified by sequencing (Hartwell Center, St. Jude Children's Research Hospital). The generation of the EDB-CAR was described previously[19]. RD114-pseudotyped retroviral particles were generated by transient transfection of 293 T cells as previously described[76].

## Generation of CAR T cells

Human peripheral blood mononuclear cells (PBMCs) were isolated using Lymphoprep (Abbott Laboratories) from de-identified elutriation chambers of leukapheresis products obtained from St. Jude's donor center or obtained from healthy donors under an IRB approved protocol at St. Jude Children's Research Hospital, after informed consent was obtained in accordance with the Declaration of Helsinki. To generate CAR T cells, we used our previously described standard protocol[76]. Briefly, PBMCs were stimulated on treated non-tissue culture 24-well plates, which were precoated with CD3 and CD28 antibodies (Miltenyi, #130-093-38, #130-093-375). Recombinant human IL-7 and IL-15 (IL-7: 10 ng/mL; IL-15: 5 ng/mL; PeproTech P13232, 40933) were added to cultures the next day. On day 2, CD3/CD28-stimulated T cells ($2.5 \times 10^5$ cells/well) were transduced with RD114-pseudotyped retroviral particles on RetroNectin (Takara)-coated plates in the presence of IL-7 and IL-15. On day 5, transduced T cells were transferred into new tissue culture 24-well plates and subsequently expanded with IL-7 and IL-15. Non-transduced (NT) T cells were prepared in the same way except for no retrovirus was added. All experiments were performed 7–14 days post-transduction using unsorted 'bulk' CAR T cells. Biological replicates were performed using PBMCs from different healthy donors.

## Flow cytometry

A FACSCanto II (BD) instrument was used to acquire flow cytometry data, which was analyzed using FlowJo v10 (FlowJo). Gating examples are shown in Supplementary Fig. 19. For surface staining of CAR T cells, samples were washed with and stained in PBS (Lonza) with 1% FBS (HyClone). For all experiments, matched isotypes or known negatives (e.g., NT T-cells, KO cell lines, known antigen-negative cell lines) served as gating controls along with positive control (e.g., anti-CD4 in all colors). LIVE/DEAD® Fixable Aqua Dead Cell Stain Kit (Invitrogen, 1:1000) or DAPI was used as a viability dye (1:10,000). T-cells were evaluated for CAR expression at multiple time points post-transduction with an anti-human IgG, F(ab')2 fragment specific-AF647; anti-mouse IgG, F(ab')2 fragment specific AF647, (Jackson ImmunoResearch 109-605-006, 115-605-006, 1:1000). Transduction was also confirmed with anti-CD19-PE (clone J3-119, Beckman Coulter, IM1285U, 0.5 μL/100 μL).

For detecting EDB expression, we used a recombinant L19 mAb as previously described[19,39], which synthesized by Thermo Fisher based on publicly available sequences, which are published[19]. Anti-COL11A1 (Invitrogen, PA5-101300) and anti-VCAN (Novus NBP2-22408) were used to detect the respective antigens. Antibodies were conjugated using Lightning-Link® Labeling Kits (Novus Bio) according to the manufacturer's instructions. Cells were prepared for surface staining at 1:300 antibody dilution based on manufactures instructions (COL11A1). All cell lines were analyzed at same voltages for each antibody in 3 technical replicates for accurate comparison. Mean of the analyses was determined and graphed accordingly.

## Co-culture assay

$1 \times 10^6$ CAR T-cells were co-cultured with $5 \times 10^5$ LM7, A673, 143B, CRL-2061, or CCL-136 tumor cells, or $3 \times 10^5$ primary fibroblasts without the provision of exogenous cytokine. CAR T-cells cultured without tumor cells served as controls. After 48 h, media was collected and frozen for later analysis. Cytokines were measured using IFNγ ELISA kits (R&D Systems, DIF50C) according to the manufacturer's instructions.

## Cytotoxicity assay

In a tissue culture-treated 96-well plate, GFP.ffluc-expressing tumor cells (12,500 A673, 143B, KO 143B, CRL2061, CCL-136, or 15,000 LM7) or 15,000 fibroblasts were co-cultured with serial dilutions of NT or CAR T cells. Each condition was plated in triplicate. After 3 days, 0.6 mg of D-luciferin (Perkin Elmer, 122799-10) was added to each well and luminescence was evaluated using an Infinite® 200 Pro MPlex plate reader (Tecan) to assess the number of viable cells in each well. Percent live tumor cells were determined by the following formula: (sample-media alone)/(tumor alone-media alone)*100.

## Xenograft mouse models

All experiments utilized 6–8 week NOD-scid IL2Rgammanull (NSG) mice obtained from St. Jude's NSG colony. Both female and male mice were utilized for intraperitoneal studies, female mice were utilized for subcutaneous study. Rodents are kept under barrier conditions in St. Jude's Animal Resource Center (ARC) to keep them specific pathogen free. A clean-to-dirty traffic pattern is used in most corridors. In all corridors, employees enter a vestibule and apply applicable PPE before entering the corridor and animal rooms to work. All cages, food, bedding and supplies are sterilized in bulk autoclaves. Rodents are maintained in microisolation caging and cage changes are performed under a change station or Class 2 A biological safety cabinet. Differential airflow is used as a preventative measure in cross contamination.

Microisolation cages (cages with filter tops or cages that fit into special ventilated racks) are used to house rodents within the facility. During 'daylight hours' animal rooms are maintained on the low-intensity white light setting. Evening hours activate a 'red light' setting. The lights in most animal rooms and corridors of the ARC are on an automated 12 h on, 12 h off light cycle. Other light cycles can be set if necessary for research objectives.

Each animal room and cubicle room in the ARC has a separate thermostat and humidistat to control temperature and humidity at the room level. Temperature and humidity are continuously monitored and alarms alert personnel to excursions from defined temperature or humidity ranges. Animal care technicians record high and low temperatures and humidity daily on a room log sheet using an electronic digital thermometer/humidistat. At endpoint (see definitions for individual models below), mice were euthanized using $CO_2$ inhalation for 3 min until breathing had stopped and there was no response to toe-pinch. Cervical dislocation followed to assure death.

**Intraperitoneal tumor models.** Mice were injected intraperitoneally (i.p.) with $1 \times 10^6$ LM7.GFP.ffLuc tumor cells, and on day 7 received a single i.v. dose of $3 \times 10^6$ T cells. For survival experiment, mice were euthanized when they reached (i) the bioluminescence Flux endpoint of $2 \times 10^{10}$ on two consecutive measurements, and (ii) they met physical euthanasia criteria (significant weight loss, signs of distress). To test for antigen loss variants, mice were injected with $1 \times 10^6$ LM7 tumor cells, and on day 7 received a single i.v. dose of $3 \times 10^6$ GFP.ffLuc-expressing T cells. Mice were euthanized at day 65 and tumors were harvested in the peritoneum for COL11A1 IHC.

**Subcutaneous tumor models.** Mice were injected subcutaneously (s.c.) with $1 \times 10^6$ A673 tumor cells in Matrigel (Corning; 1:1 diluted in PBS). On day 7, mice received a single i.v. dose of $1 \times 10^6$ T cells via tail vein injection. Tumor growth was assessed by serial caliper measurements from a third-party animal technician to allow for a blinded study. Mice were euthanized when (i) they met physical euthanasia criteria (significant weight loss, signs of distress), (ii) the tumor burden was ~4000 mm³, or reached a radiance of $\geq 1 \times 10^{10}$ for 10 days, or (iii) recommended by St. Jude veterinary staff; maximum tumor size burden was not exceeded.

## Bioluminescence imaging

Mice were imaged as described previously[19]. Briefly, they were injected i.p. with 150 mg/kg of D-luciferin 5–10 min before imaging, anesthetized an induction chamber (2–3% isoflurane, with oxygen), after which placed in the imaging instrument and fitted with a nose cone connected to a vaporizer to maintain isoflurane (1.0–2.5%) during the procedure. Images were acquired on a Xenogen IVIS-200 imaging system. The photons emitted from the luciferase-expressing tumor cells were quantified using Living Image software (Caliper Life Sciences).

## Statistical analysis

All experiments were performed at least in triplicates. For comparison between two groups, two-tailed t-test was used. For comparisons of three or more groups, values were log transformed as needed and analyzed by ANOVA with Tukey's post-test. Survival was analyzed by Kaplan–Meier method and by the log-rank test. Statistical analyses were conducted with Prism software (Version 9.0.0, GraphPad Software).

## Reagent and protocol availability

Contact Jinghui Zhang at jinghui.zhang@stjude.org or Stephen Gottschalk at stephen.gottschalk@stjude.org.

## Reporting summary

Further information on research design is available in the Nature Portfolio Reporting Summary linked to this article.

## Data availability

The raw RNA-seq publicly available data for PCGP and St Jude ClinGen samples are available on St Jude Cloud Genomics Platform (https://platform.stjude.cloud/data/cohorts/pediatric-cancer) under the accessions SJC-DS-1001, SJC-DS-1003, SJC-DS-1004 and SJC-DS-1007. The publicly available NCI TARGET data are available in dbGaP under accession phs000218. NCI TARGET data used in this study are available in dbGaP under accession phs000218 https://www.ncbi.nlm.nih.gov/projects/gap/cgi-bin/study.cgi?study_id=phs000218.v1.p1. The publicly available GTEx RNAseq data used in this study can be accessed through the dbGAP accession phs000424.v8.p2 https://www.ncbi.nlm.nih.gov/projects/gap/cgi-bin/study.cgi?study_id=phs000424.v8.p2. The Iso-seq data used for verifying alternative splicing of FN1, TNC, COL6A3 in osteosarcoma can be accessed in the European Genome-phenome Archive (EGA) under accession number EGAS00001007766. The publicly available GTEx proteomics data used in this study can be accessed through PXD016999. The processed publicly available pediatric brain tumor proteomics data used in this study can be accessed through the NCI proteomics data commons https://pdc.cancer.gov/pdc/study/PDC000432. PDX IDs and their associated accessions can be found at Supplementary Data 3. The remaining data are available within the Article, Supplementary Information or Source Data file. Source data are provided with this paper.

## Code availability

RNA-seq data were processed by a custom pipeline (https://github.com/shawlab-moffitt/CSEminer-manuscript). The data processing code and data can be accessed through Zenodo https://zenodo.org/records/10672928 and https://zenodo.org/records/10594740.

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

## Acknowledgements

The authors would like to thank the research staff of St. Jude Centers and Cores (listed under 'Funding'), who assisted in the conduct of the experiments, and members of the CPTAC consortium. Figure 1A was

created in part with BioRender (Biorender.com), for which we have a license. We thank Jiyang Yu, Paulina Velasquez, Giedre Krenciute, Christopher DeRenzo, and members of Jinghui Zhang's lab for the helpful discussion. We thank Delaram Becksfort for helping with the splice variant analysis. We thank Gang Wu, Shibiao Wan, and Yawei Hui from the Center for Applied Bioinformatics for their help with the GTEx mapping, and Jobin Sunny from the Department of Computational Biology with sequence data deposition. We thank Elizabeth Stewart, Hong Wang, Mingming Niu, Junmin Peng, and Yuxin Li for their help on the St. Jude Proteomics data. We thank Delaram Becksfort for her help with the manuscript revision. This work was supported by the Alex's Lemonade Stand Foundation and Cure4Cam Foundation (ALSF; Young Investigator Grant; J.W.), National Institutes of Health (NIH) grant 1F31CA257757 (E.W.), R01CA216391 (J.Z.), NIH P30CA076292 (T.S.), the Alliance for Cancer Gene Therapy (S.G.), St. Jude's Translational Immunology and Immunotherapy initiative (J.Z., S.G.), and the American Lebanese Syrian Associated Charites (J.Z., S.G.), Moffit Quantitative Science Team Science Grant and Bio2 Department Pilot (T.S.), and the Florida Department of Health Live Like Bella Pediatric Research Initiative (T.S.). Animal imaging was performed by the Center for In Vivo Imaging and Therapeutics, which is supported in part by the National Cancer Institute (NCI) grant P30CA021765. Gene editing of cell lines was performed by the Center for Advanced Genome Engineering, which is supported in part by NCI P30CA021765. The content is solely the responsibility of the authors and does not necessarily represent the official views of the NIH.

## Author contributions

Conceptualization: T.S., J.Z., S.G.; Methodology: T.S., J.W., L.T., E.W.; Investigation: T.S., J.W., L.T., E.W., S.P., R.P., Jian W., S.C.K., M.L., H.S., Y.F., F.O., C.C.L., X.Z., J.Z., S.G.; Resources: J.Z., S.G.; Formal analysis: T.S., J.W., L.T., E.W., S.C.K., H.S., J.Z., S.G.; Supervision: J.Z., S.G.; Funding acquisition: J.W., E.W., J.Z., S.G.; Writing – original draft preparation: T.S., J.W., L.T., J.Z., S.G.; Writing – review and editing: T.S., J.W., L.T., E.W., R.P., Jian W., R.P., S.C.K., M.L., H.S., Y.F., F.O., C.C.L., X.Z., J.Z., S.G.

## Competing interests

T.S., J.W., E.W., J.Z., and S.G. have patent applications in the fields of T-cell and gene therapy for cancer. The following two patent applications are directly related to the manuscript: (i) Chimeric antigen receptors targeting splice variants of the extracellular matrix proteins tenascin C (TNC) and procollagen 11A1 (COL11A1), WO/2022/147075, PCT/ US2021/065445 (Inventors: S.G., J.W., E.W., T.S., J.Z.; Institution: St. Jude Children's Research Hospital), and (ii) Chimeric antigen receptors for direct and indirect targeting of fibronectin-positive tumors, WO/2021/ 016091, No.PCT/US2021/065445 (Inventors: S.G., J.W., T.S., J.Z.; Institution: St. Jude Children's Research Hospital). S.G. is a member of the Scientific Advisory Board of Be Biopharma and CARGO, and the Data and Safety Monitoring Board (DSMB) of Immatics and has received honoraria from TESSA Therapeutics within the last year. The remaining authors declare no competing interests.
