## [Peer Review File · Nature Communications]

Discovery of immunotherapy targets for pediatric solid and brain tumors by exon-level expressionEditorial Note: This manuscript has been previously reviewed at another journal that is not operating a transparent peer review scheme. This document only contains reviewer comments and rebuttal letters for versions considered at Nature Communications.

Reviewers' Comments:

Reviewer #1:

Remarks to the Author:

The authors have largely addressed the main points I raised. They have clarified what is meant by "Cancer Specific Exons" to indicate that includes cancer over-expressed exons and further described their potential origin as a result of transcriptional deregulation. This will likely help readers understand the nature of the work.

I do want to explore one element of the author's response which I may have misunderstood. In their second point, the authors describe their appropriate and validated processes for the development of the CARs and the demonstration of the specificity of these. My question was not so much this but pertained to a comment in the "Validation of cell surface expression of selected CSEs in patient-derived xenograft (PDX) models" section of the paper. In this section, the authors stated "For VCAN1 and EDB, we took advantage of exon-specific mAbs matching the CSEs and performed flow cytometric analysis of 15 pediatric PDX samples (5 OS, 5 EWS, 5 RMS; Supplementary Table3)." The question raised in the review was more about how the specificity of these antibodies for the isoforms of the genes which include the CSE was established. I took this statement and figure 3 to mean that these mAb were specific to the proteins expressed from isoforms of the genes that contained cancer specific exons but would not bind those isoforms in which the CSE are not expressed. If the interpretation of the statement is incorrect and a misinterpretation of the author's intent, then it possible that other readers might also make the same mistake. Clarifying this may be helpful to readers.

Reviewer #2:

Remarks to the Author:

I have reviewed the extensive revisions by the authors. In general, I am supportive of publication. But there is one thing I think should be updated. We requested IHC showing expression of COL11A on primary tumor samples, which the authors provided. However, they only provide a single example of high, a single example of low, and a single example of negative. This does not allow the reader to understand the heterogeneity of the target. I doubt the authors stained only 3 samples. I think they should report staining results for an unbiased collection of samples. I do not accept the PDX data as a proxy given the shift of PDX expression over time as it gets passaged in a mouse.

Reviewer #3:

Remarks to the Author:

Overall, the authors do a nice job responding to the initial reviewer comments. Below, we have some additional comments and suggestions.

Minor: "407587_0_data_set_8307717_s432dr.pdf" seems to include supplemental (not Extended data figures), so this may have been an older version and made review somewhat confusing since they did not match up to the text. It seems that the Extended data figures are in the full version of the text.

AUTHOR RESPONSE:

We thanked the reviewer for this suggestion. However, the definition of tumor specificity used in the Nat. Comm. Paper (Rivero-Hinojosa et al, 2021) cited by the reviewer is quite different from what we used here. In the Nat. Comm. paper, the neoantigen was based on somatic SNVs, indels, gene fusion and novel splice junctions while we focused only on the known transcript and our target list is derived based on differential expression of tumor versus normal samples. As stated in our response to the previous question, we only have 9 AS targets, 8 of which can be validated by a simple blat query using the peptide sequence. As the remaining targets are gene-based, a gene-level validation using proteomics is sufficient and there is no need to create a tumor-specific custom database. In the revised Discussion, as a response to the preceding question, we also addressed the need to use proteomics databases for validating novel isoforms for future studies.

Reviewer response:

Since most of the targets presented here are expression-based with very rare AS events (unvalidated), the authors should re-word the main description on the portal website. The way it is written is mis-leading- using the term "splicing-derived" is overstated and inaccurate. We suggest reword to: "Thus, we developed Cancer-Specific Isoform Miner (CSE-Miner), a pipeline for mining exon-based CAR-T targets".

The current description on the main site is: "Immunotherapy with T-cells expressing chimeric antigen receptors (CARs) holds the promise to improve outcomes for children with solid and brain tumors. However, at present, there is a limited array of targetable antigens. We posit here that targets generated by alternative splicing events in cancer cells present an untapped class of CAR targets since many of these are tumor-specific and overexpressed in malignant cells. Thus, we developed Cancer-Specific Isoform Miner (CSE-Miner), a pipeline for mining splicing derived CAR-T targets."

AUTHOR RESPONSE:

We thank the reviewer for the suggestion of comparing with the published algorithms. We feel that SNAF is not an appropriate tool as it was designed for finding splicing neoantigen, which is different from CAR-T targets.

We ran IRIS (<https://github.com/Xinglab/IRIS> PMID: 37192158) on 136 osteosarcoma samples included in our study, which quantify alternative splicing through percent splice in (PSI) by rMATS, which is part of the IRIS package. Note, IRIS only processes skipped exon (SE) but does not evaluate alternative 5'/3' splicing or intron retention events. Altogether, IRIS identified 112 alternative splicing CAR-T targets; however, none matched the three validated CSE targets in FN1/TNC/COL6A3 were identified by CSE for osteosarcoma.

Reviewer response:

The above statement that IRIS does not evaluate 5'/3' start site or RI events is not correct. From the IRIS manuscript, "To quantify AS events, IRIS extracts PSI values (29) for all AS events in the rMATS-turbo output file and read counts for all SJs in the RNA-seq alignment file. To remove low-confidence PSI estimates, AS events with low RNA-seq coverage, defined as events with an average read count of less than 10 for the sum of all corresponding SJs in a given sample or sample group (e.g., tumor or normal tissue type), are masked as having missing values in the output file of IRIS-characterized AS events. This pipeline discovers and quantifies all major types of AS events, including SE, A5SS, A3SS, and RI events".

AUTHOR RESPONSE:

We speculate that the absence of these targets by IRIS might be related to the incompleteness of extracellular domain database used by IRIS for filtering the CAR-T targets as they were in the

intermediate file but not the final output. Given that there is well-established experimental evidence of FN1 EDB CAR-T cells in killing osteosarcoma cells (PMID: 33355188) while the experimental data shown in our manuscript and prior publication show that FN1, TNC and COL6A3 are well-known extracellular matrix proteins (PMID: 20690820, PMID: 36102918, PMID: 21691338), we consider these misses are false negatives in the IRIS pipeline.

Reviewer response:

Did the authors add their list of genes with validated extracellular domains to the IRIS database and rerun to confirm this is truly the reason for the false negatives? This would be a simple check, rather than postulating here.

AUTHOR RESPONSE:

As the accuracy of IRIS is out of scope of the present study, we hope to focus on presenting the discovery potential of CSEminer instead of investigating the issues or accuracies with previously published tools.

Reviewer response:

It is imperative in a response such as this to report how algorithms were run, as parameters and filtering have a significant effect on the results. For example, what were the cut-offs used for read counts, dPSIs, and pval/FDR and similarly, what parameters were supplied when running the tool(s)? Furthermore, did the non-validated candidates match (in the intermediate or final results)? What total percentage did match? This will enable reviewers familiar with the algorithms' inner workings to evaluate whether the analysis was performed in a sound manner and allow suggestions for improving the analysis/adding some checks. Can the authors summarize this?

Within the splice variant analysis section, the authors state: "To find splice variants, we queried the tumor variant files on the St Jude Cloud for those located within 10bp of splice acceptor and donor sites of the 9 AS exons in COL6A3 (chr2:237378636- 237379235), FN1 (chr2:215392931-215393203), POSTN (chr13:37574572-37574652), TNC (chr9:115048260-115048532), VCAN(chr5:83519349-83522309), NRCAM (chr7:108191254- 108191283), FYN (chr6:111699515-111699670), PICALM (chr11:85990250-85990378), and CLSTN1 (chr1:9756481-9756510). No variants were found for COL6A3, POSTN, TNC, NRCAM, FYN, PICALM, and CLSTN1."

Can the authors clarify that this was indeed a splice variant analysis and not a somatic mutation analysis? From the read above, it sounds like the authors searched the DNA data for somatic mutations near splice acceptor and donor sites, but this could be easily confused with a true splice event/variant analysis using RNA expression data, and not DNA variant data. If this analysis refers to somatic mutations affecting splicing, then the heading should be reworded to reflect this for clarity.

AUTHOR RESPONSE: We only included known/validated exons from Gencode v31.

Reviewer response:

Please be more specific, as there are different gffs/gtfs for v31. For example, was the "Basic gene annotation", "Comprehensive gene annotation", etc. utilized here? Method details are important because exon boundaries and number of exons change depending on what file is used.

AUTHOR RESPONSE:

"-m union --nonunique all" for htseq-count was used, as documented in the origin manuscript. Therefore, if a read spanned a splice junction, it would be counted for both exons. For small exons, this could potentially lead to overestimation of expression level. To overcome this issue, at the step of calculating FPKM, we used exon length as the read length when the length of an exon was shorter

than the read length. Additionally, as this issue affects both the tumor and the normal samples, the tumor-versus-normal differential expression analysis used to identify CSE can also mitigate this effect. To clarify this point, we have modified the original writing in

AUTHOR RESPONSE:

We thank the reviewer for their thoughtful comment. Our pipeline does not perform the task of assigning each CSE to a specific transcript. This is because antigen for CAR-T cells is based on exon-specific peptide (antibody), and the presence of the exon in one or multiple transcripts will not affect antibody specificity. For example, based on the transcript annotation in gene code, the extra domain B fibronectin (EDB) should be encoded by three isoforms that differ in transcription of exons downstream. Given that the protein products of the transcripts encode EDB fibronectins that are only present in the ECM but not in the plasma, the relative quantity of these transcripts do not affect the antitumor activity of EDB as described in our prior publication (PMID 33355188)

Reviewer response:

This is not entirely true. The sequence may be identical, but downstream and upstream exons may impact the way the translated protein is modified, leading to possible structural changes. This would be of particular concern if there are sites of disulfide bonding, as these could make the sequences or sites of interest inaccessible, especially if exons are small. Although it may not affect the design of the CAR, this is a limitation that would be worthwhile to mention.

Point-By-Point Response

Reviewer #1 (Remarks to the Author):

The authors have largely addressed the main points I raised. They have clarified what is meant by "Cancer Specific Exons" to indicate that includes cancer over-expressed exons and further described their potential origin as a result of transcriptional deregulation. This will likely help readers understand the nature of the work.

I do want to explore one element of the author's response which I may have misunderstood. In their second point, the authors describe their appropriate and validated processes for the development of the CARs and the demonstration of the specificity of these. My question was not so much this but pertained to a comment in the "Validation of cell surface expression of selected CSEs in patient-derived xenograft (PDX) models" section of the paper. In this section, the authors stated "For VCAN1 and EDB, we took advantage of exon-specific mAbs matching the CSEs and performed flow cytometric analysis of 15 pediatric PDX samples (5 OS, 5 EWS, 5 RMS; Supplementary Table 3)." The question raised in the review was more about how the specificity of these antibodies for the isoforms of the genes which include the CSE was established. I took this statement and figure 3 to mean that these mAb were specific to the proteins expressed from isoforms of the genes that contained cancer specific exons but would not bind those isoforms in which the CSE are not expressed. If the interpretation of the statement is incorrect and a misinterpretation of the author's intent, then it possible that other readers might also make the same mistake. Clarifying this may be helpful to readers.

RESPONSE: The reviewer correctly interpreted the mAB specificity to CSE. We modified the sentence as follows: "For VCAN1 and EDB, we took advantage of mAbs that recognize the part of the molecule that is encoded by the differentially expressed exon and performed flow cytometric analysis of 15 pediatric PDX samples (5 OS, 5 EWS, 5 RMS; Supplementary Table 3)." We hope that this adds clarity - we are grateful for the reviewer's effort to clarify this issue with us.

Reviewer #2 (Remarks to the Author):

I have reviewed the extensive revisions by the authors. In general, I am supportive of publication. But there is one thing I think should be updated. We requested IHC showing expression of COL11A on primary tumor samples, which the authors provided. However, they only provide a single example of high, a single example of low, and a single example of negative. This does not allow the reader to understand the heterogeneity of the target. I doubt the authors stained only 3 samples. I think they should report staining results for an unbiased collection of samples. I do not accept the PDX data as a proxy given the shift of PDX expression over time as it gets passaged in a mouse.

RESPONSE: We would like to thank the reviewer for their comment and clarify that we had analyzed 18 OS, 11 EWS, and 37 RMS tumor samples as stated in the text ‘*Of the primary tumors, 12/18 OS, 7/11 EWS, and 14/37 RMS samples highly expressed Col11A1 (Extended Data Fig 5); 5/18 OS, 3/11 EWS, and 10/37 RMS tumor samples showed low expression, respectively.*’

We have now increased the number to ten individual tumors per histology in **Extended Data Fig 5** and have clarified the figure legend.

Reviewer #3 (Remarks to the Author):

Overall, the authors do a nice job responding to the initial reviewer comments. Below, we have some additional comments and suggestions.

RESPONSE: Thank you.

Minor: “407587_0_data_set_8307717_s432dr.pdf” seems to include supplemental (not Extended data figures), so this may have been an older version and made review somewhat confusing since they did not match up to the text. It seems that the Extended data figures are in the full version of the text.

RESPONSE: Thank you for your comment. With the previous submission, the Extended Data Figures were included in the main manuscript. We confirm that ‘407587_0_data_set_8307717_s432dr.pdf’ was the converted Supplementary Figure file. We have clearly labeled all files with the resubmission. Of note, following our past experience of publishing in Nature journals (e.g. Nature, Nature Medicine), Extended Data Figures shall appear in the online version of the manuscript and should not be part of the Supplementary Information. Supplementary figures, on the other hand, do appear in Supplementary Information.

AUTHOR RESPONSE:

We thanked the reviewer for this suggestion. However, the definition of tumor specificity used in the Nat. Comm. Paper (Rivero-Hinojosa et al, 2021) cited by the reviewer is quite different from what we used here. In the Nat. Comm. paper, the neoantigen was based on somatic SNVs, indels, gene fusion and novel splice junctions while we focused only on the known transcript and our target list is derived based on differential expression of tumor versus normal samples. As stated in our response to the previous question, we only have 9 AS targets, 8 of which can be validated by a simple blat query using the peptide sequence. As the remaining targets are gene-based, a gene-level validation using proteomics is sufficient and there is no need to create a tumor-specific custom database. In the revised Discussion, as a response to the preceding question, we also addressed the need to use proteomics databases for validating novel isoforms for future studies.

Reviewer response:

Since most of the targets presented here are expression-based with very rare AS events (unvalidated), the authors should re-word the main description on the portal website. The way it is written is mis-leading– using the term “splicing-derived” is overstated and inaccurate. We suggest reword to: “Thus, we developed Cancer-Specific Isoform Miner (CSE-Miner), a pipeline for mining exon-based CAR-T targets”.

RESPONSE: The AS events reported in the study, though accounting for only a small fraction of all the CAR-T targets (9 out of 157) and the expression of their corresponding peptides have been validated by proteomics data. This is documented in the section “Curation of expression specificity and splicing pattern of CSE targets” in Methods. Out of the three targets selected for validation of cell surface expression (FN1, VCAN1, COL11A1), two were AS targets (FN1 and VCAN1). In the revised manuscript, we have provided further evidence of isoform expression using full-length RNA-seq data generated from the matching tumor types. Therefore, the term “unvalidated” used by the reviewer to describe our AS targets is not appropriate.

We appreciate the reviewer's careful check on the web page and have modified the text to "Thus, we developed Cancer-Specific Exon Miner (CSE-Miner), a pipeline for mining exon-based CAR-T targets". Please see a screenshot of the updated web page.

The current description on the main site is: "Immunotherapy with T-cells expressing chimeric antigen receptors (CARs) holds the promise to improve outcomes for children with solid and brain tumors. However, at present, there is a limited array of targetable antigens. We posit here that targets generated by alternative splicing events in cancer cells present an untapped class of CAR targets since many of these are tumor-specific and overexpressed in malignant cells. Thus, we developed Cancer-Specific Isoform Miner (CSE-Miner), a pipeline for mining splicing derived CAR-T targets."

AUTHOR RESPONSE:

We thank the reviewer for the suggestion of comparing with the published algorithms. We feel that SNAF is not an appropriate tool as it was designed for finding splicing neoantigen, which is different from CAR-T targets.

We ran IRIS (<https://github.com/Xinglab/IRIS> PMID: 37192158) on 136 osteosarcoma samples included in our study, which quantify alternative splicing through percent splice in (PSI) by rMATS, which is part of the IRIS package. Note, IRIS only processes skipped exon (SE) but does not evaluate alternative 5'/3' splicing or intron retention events. Altogether, IRIS identified 112 alternative splicing CAR-T targets; however, none matched the three validated CSE targets in FN1/TNC/COL6A3 were identified by CSE for osteosarcoma.

Reviewer response:

The above statement that IRIS does not evaluate 5'/3' start site or RI events is not correct. From the IRIS manuscript, "To quantify AS events, IRIS extracts PSI values (29) for all AS events in the rMATS-turbo output file and read counts for all SJs in the RNA-seq alignment file. To remove low-confidence PSI estimates, AS events with low RNA-seq coverage, defined as events with an average read count of less than 10 for the sum of all corresponding SJs in a given sample or sample group (e.g., tumor or normal tissue type), are masked as having missing values in the output file of IRIS-characterized AS events. This pipeline discovers and quantifies all major types of AS events, including SE, A5SS, A3SS, and RI events".

RESPONSE: We apologize for misinterpreting the isoform characterization function of IRIS and added the following statement to the Discussion of the manuscript to highlight that use of

alternative methods such as IRIS, coupled with extensive curation of the database and proteomics validation, can potentially lead to additional targets in the pediatric cancer data set. New text was added (underlined) to the Discussion:

“Future studies that incorporate novel isoform discovery with CSE analysis or other newly published methods such as Isoform peptides from RNA splicing for Immunotherapy target Screening (IRIS) (PMID 37192158) followed by curation and validation using proteomics databases may further expand the repertoire of AS targets.”

AUTHOR RESPONSE:

We speculate that the absence of these targets by IRIS might be related to the incompleteness of extracellular domain database used by IRIS for filtering the CAR-T targets as they were in the intermediate file but not the final output. Given that there is well-established experimental evidence of FN1 EDB CAR-T cells in killing osteosarcoma cells (PMID: 33355188) while the experimental data shown in our manuscript and prior publication show that FN1, TNC and COL6A3 are well-known extracellular matrix proteins (PMID: 20690820, PMID: 36102918, PMID: 21691338), we consider these misses are false negatives in the IRIS pipeline.

Reviewer response:

Did the authors add their list of genes with validated extracellular domains to the IRIS database and rerun to confirm this is truly the reason for the false negatives? This would be a simple check, rather than postulating here.

RESPONSE:

To verify that our interpretation of the false negative result of IRIS is correct, we manually added the three genes with AS exons by incorporating the exons that encode the validated extracellular domains IRIS' gene extracellular annotation database and then rerun IRIS using this updated file. The results from the rerun now include the 3 AS exons. This exercise confirms our interpretation of the false negative results of IRIS.

New information manually added to represent validated extracellular domains of FN1 TNC COL6A3 in IRIS' gene extracellular annotation database (hg19 coordinate):

ENSG00000041982:TNC:chr9:-:117810538:117810811:117808961:117835881

ENSG00000115414:FN1:chr2:-:216257653:216257926:216256537:216259250

ENSG00000163359:COL6A3:chr2:-:238287278:238287878:238285987:238289557

AUTHOR RESPONSE:

As the accuracy of IRIS is out of scope of the present study, we hope to focus on presenting the discovery potential of CSEminer instead of investigating the issues or accuracies with previously published tools.

Reviewer response

It is imperative in a response such as this to report how algorithms were run, as parameters and filtering have a significant effect on the results. For example, what were the cut-offs used for read counts, dPSIs, and pval/FDR and similarly, what parameters were supplied when running the tool(s)? Furthermore, did the non-validated candidates match (in the intermediate or final results)? What total percentage did match? This will enable reviewers familiar with the algorithms' inner workings to evaluate whether the analysis was performed in a sound manner and allow suggestions for improving the analysis/adding some checks. Can the authors summarize this?

RESPONSE: Our study did NOT use IRIS for data analysis to generate CAR-T targets. Our test analysis on osteosarcoma, performed in the last revision, used the parameters defined in the IRIS parameter file https://github.com/Xinglab/IRIS/blob/master/example/NEPC_test.para. We ran the analysis as a response to the reviewer's inquiry and the results from IRIS are incorporated into the manuscript. We want to emphasize that identifying ASE targets is only one of the first step involved in this study and the downstream curation/validation has taken 3 years to complete. Therefore, tuning of IRIS, rationalizing all the parameters used, comparing the results with validated versus unvalidated targets is needed only for a new research project that uses IRIS for finding CAR-T targets. We have encouraged such pursuit in our response to the preceding question by expanding our discussion to show tools such as IRIS can be useful for finding novel isoforms.

Within the splice variant analysis section, the authors state: "To find splice variants, we queried the tumor variant files on the St Jude Cloud for those located within 10bp of splice acceptor and donor sites of the 9 AS exons in COL6A3 (chr2:237378636- 237379235), FN1 (chr2:215392931-215393203), POSTN (chr13:37574572-37574652), TNC (chr9:115048260-115048532), VCAN(chr5:83519349-83522309), NRCAM (chr7:108191254- 108191283), FYN (chr6:111699515-111699670), PICALM (chr11:85990250-85990378), and CLSTN1 (chr1:9756481-9756510). No variants were found for COL6A3, POSTN, TNC, NRCAM, FYN, PICALM, and CLSTN1."

Can the authors clarify that this was indeed a splice variant analysis and not a somatic mutation analysis? From the read above, it sounds like the authors searched the DNA data for somatic mutations near splice acceptor and donor sites, but this could be easily confused with a true splice event/variant analysis using RNA expression data, and not DNA variant data. If this analysis refers to somatic mutations affecting splicing, then the heading should be reworded to reflect this for clarity.

RESPONSE: The variants present in a tumor sample include both somatic and germline variants. We further clarified this by explicitly stating (underlined) the following in the Methods Section (page 2): *"To find splice variants, we queried the tumor variant files which contain both somatic and germline variants for those located within"* This analysis was included in revision as a response to a question raised by the reviewer in the previous review related to the DNA variants in an article on Human Genome Variation Journal. *"Were any of the high-priority splice variants previously discovered in any other cancers or diseases? Additionally, what is known about the genomic landscape of the models used – are there any mutations predicted to be responsible for the splice events (eg: <https://www.nature.com/articles/hgv201543>)?"*

Given the absence of such candidate variants in regions involved in splicing, there is no need to further investigate disruption of splicing caused by such variants.

AUTHOR RESPONSE: *We only included known/validated exons from Gencode v31.*

Reviewer response:

Please be more specific, as there are different gffs/gtfs for v31. For example, was the "Basic gene annotation", "Comprehensive gene annotation", etc. utilized here? Method details are important because exon boundaries and number of exons change depending on what file is used.

RESPONSE: We used human comprehensive gencode v31 primary assembly gene annotation. We have updated the information in the Methods section.

AUTHOR RESPONSE:

“-m union --nonunique all” for htseq-count was used, as documented in the origin manuscript. Therefore, if a read spanned a splice junction, it would be counted for both exons. For small exons, this could potentially lead to overestimation of expression level. To overcome this issue, at the step of calculating FPKM, we used exon length as the read length when the length of an exon was shorter than the read length. Additionally, as this issue affects both the tumor and the normal samples, the tumor-versus-normal differential expression analysis used to identify CSE can also mitigate this effect. To clarify this point, we have modified the original writing in

AUTHOR RESPONSE:

We thank the reviewer for their thoughtful comment. Our pipeline does not perform the task of assigning each CSE to a specific transcript. This is because antigen for CAR-T cells is based on exon-specific peptide (antibody), and the presence of the exon in one or multiple transcripts will not affect antibody specificity. For example, based on the transcript annotation in gene code, the extra domain B fibronectin (EDB) should be encoded by three isoforms that differ in transcription of exons downstream. Given that the protein products of the transcripts encode EDB fibronectins that are only present in the ECM but not in the plasma, the relative quantity of these transcripts do not affect the antitumor activity of EDB as described in our prior publication (PMID 33355188)

Reviewer response:

This is not entirely true. The sequence may be identical, but downstream and upstream exons may impact the way the translated protein is modified, leading to possible structural changes. This would be of particular concern if there are sites of disulfide bonding, as these could make the sequences or sites of interest inaccessible, especially if exons are small. Although it may not affect the design of the CAR, this is a limitation that would be worthwhile to mention.

RESPONSE: The reviewer raises an interesting point, and we had already stated in the discussion:

Validation of consistent expression of target genes in individual tumor cells is critical, which may involve performing IHC of primary patient samples and evaluating gene expression at single cell level by re-analyzing appropriate scRNA-seq data set as demonstrated in our validation of COL11A1.

We have now modified the sentence:

Validation of consistent expression of target genes and to exclude epitope masking due to the tertiary structure of the protein in individual tumor cells is critical, which may involve performing IHC of primary patient samples and evaluating gene expression at single cell level by re-analyzing appropriate scRNA-seq data set as demonstrated in our validation of COL11A1.

Reviewers' Comments:

Reviewer #1:

Remarks to the Author:

The authors have addressed my final enquiry

Reviewer #2:

Remarks to the Author:

Thank you to the authors for updating the IHC - it really increases the value of this manuscript to the field!

Reviewer #3:

Remarks to the Author:

All of our comments have been adequately addressed, but if Nature is taking the stance of all work being reproducible, code reproducibility should be addressed.

Reviewer #4:

Remarks to the Author:

I co-reviewed this manuscript with one of the other main reviewers.

POINT-BY-POINT RESPONSE

Reviewer #1 (Remarks to the Author):

The authors have addressed my final enquiry.

RESPONSE: Thank you.

Reviewer #1 (Remarks on code availability):

I do not have sufficient expertise to analyse the code provided.

RESPONSE: n/a.

Reviewer #2 (Remarks to the Author):

Thank you to the authors for updating the IHC - it really increases the value of this manuscript to the field!

RESPONSE: Thank you.

Reviewer #3 (Remarks to the Author):

All of our comments have been adequately addressed, but if Nature is taking the stance of all work being reproducible, code reproducibility should be addressed.

RESPONSE: Thank you.

Reviewer #3 (Remarks on code availability):

We reviewed the initial code which was St. Jude HPC specific and not reproducible by any outsider. The authors had created a new repository to store the code. As it stands, the code is "example" code for re-running analyses and in some instances contains local paths so it is not "ready to use", but more of a guide. A better way to do this would be to containerize the code so anyone can use it. The figure generation folder is potentially reproducible, but we did not test it. There is no code available for the CSE-Miner application. The README is not detailed enough with instructions for a user to run the code.

RESPONSE: We provided the following comments also in the 'Editorial Request Document':

As recommended by the reviewer, we have implemented a docker image of the Step 3 figure data generation process, which is provided on Zenodo. <https://zenodo.org/records/10594740>.

We have added a Shiny app of the exon heatmap shown in Figures 2 and Extended Figures 1-3. <https://zenodo.org/records/10600281> with a companion Shinyapps.io live on <https://shawlab-moffitt.shinyapps.io/interactiveshinyappexonheatmapstjudepedcancer/>

From the GitHub page for 3_figure_data_generation https://github.com/shawlab-moffitt/CSEminer-manuscript/tree/main/3_figure_data_generationwith, we have simplified the process so the user would not need to download data from dropbox.

In addition to the presented heatmap figure, an existing web interface is also available to explore the prioritized targets along with other targets using our Shiny app located in cseminer.stjude.org.

Our bioinformatics interns were able to run both the Shiny apps and the Dockerized code in their computing environment. We recognize that there will always be technical challenges associated with executing software code due to differences in user expertise. We hope that by providing both the Dockerized environment and Shiny apps will prevent future issues. Regardless, we have included a note in the GitHub page and Zenodo repositories that any installation questions can be addressed to timothy.shaw@moffitt.org or liqing.tian@stjude.org.

The entire Github code from <https://github.com/shawlab-moffitt/CSEminer-manuscript/> has also been deposited in Zenodo <https://zenodo.org/doi/10.5281/zenodo.1067292>

We have also uploaded the supplementary data to Zenodo:

<https://zenodo.org/records/10607084>

<https://zenodo.org/records/10607288>

<https://zenodo.org/records/10783179>

<https://zenodo.org/records/10783329>

<https://zenodo.org/records/10783398>.

Note that the complete exon quantification is over 27.6G and was too large to get uploaded to Zenodo; therefore, a smaller example data is provided <https://zenodo.org/records/10783329>.